# Split Intein-Mediated Protein Ligation for detecting protein-protein interactions and their inhibition

Zhong Yao[1], Farzaneh Aboualizadeh[1], Jason Kroll [2], Indira Akula[1], Jamie Snider[1], Anna Lyakisheva[1], Priscilla Tang[1], Max Kotlyar[3], Igor Jurisica[3,4,5,6], Mike Boxem [2] & Igor Stagljar [1,7,8,9]✉

Here, to overcome many limitations accompanying current available methods to detect protein-protein interactions (PPIs), we develop a live cell method called Split Intein-Mediated Protein Ligation (SIMPL). In this approach, bait and prey proteins are respectively fused to an intein N-terminal fragment (IN) and C-terminal fragment (IC) derived from a re-engineered split intein GP41-1. The bait/prey binding reconstitutes the intein, which splices the bait and prey peptides into a single intact protein that can be detected by regular protein detection methods such as Western blot analysis and ELISA, serving as readouts of PPIs. The method is robust and can be applied not only in mammalian cell lines but in animal models such as *C. elegans*. SIMPL demonstrates high sensitivity and specificity, and enables exploration of PPIs in different cellular compartments and tracking of kinetic interactions. Additionally, we establish a SIMPL ELISA platform that enables high-throughput screening of PPIs and their inhibitors.

[1] Donnelly Centre, University of Toronto, Toronto, ON, Canada. [2] Division of Developmental Biology, Institute of Biodynamics and Biocomplexity, Faculty of Science, Utrecht University, Utrecht, Netherlands. [3] Krembil Research Institute, University Health Network, Toronto, ON, Canada. [4] Department of Medical Biophysics, University of Toronto, Toronto, ON, Canada. [5] Department of Computer Science, University of Toronto, Toronto, ON, Canada. [6] Institute of Neuroimmunology, Slovak Academy of Sciences, Bratislava, Slovak Republic. [7] Department of Biochemistry, University of Toronto, Toronto, ON, Canada. [8] Department of Molecular Genetics, University of Toronto, Toronto, ON, Canada. [9] Mediterranean Institute for Life Sciences, Meštrovićevo Šetalište 45, HR-21000 Split, Croatia. ✉email: igor.stagljar@utoronto.ca

Protein–protein interactions (PPIs) compose many fundamental steps in most biological processes[1]. Thus, PPI detection and analysis are essential for understanding molecular mechanisms of biological processes, elucidating mechanistic details of disease occurrence and progression, as well as for developing new treatments and diagnostics.

Numerous methods have been developed to detect PPIs[1,2]. However, based on the underpinning principles of their design, these methods present bias for certain sets of PPIs and are accompanied by limitations[3]. For example, stable PPIs are often easily monitored by various methods, but transient and weak PPIs, which play essential regulatory roles, are far more difficult to detect. PPIs occurring in some special cellular locations can also only be examined with specific methods. Additionally, some approaches reconstitute PPIs in model organisms such as yeast[4], which may not accurately represent native physiological environments, while methods based on affinity purification are usually biased by overrepresentation of abundant proteins and underrepresentation of weak PPIs that are easily lost during purification. Many methods also suffer from low quantifiability or throughput.

In this study, we develop a method for PPI detection called Split-Intein-Mediated Protein Ligation (SIMPL). In this approach, a split intein is used as a sensor for protein interactions. An intein is a protein fragment possessing enzymatic activity which allows it to excise itself from its parental peptide while ligating (via formation of a peptide bond) the flanking protein regions (referred to as the N-terminal extein (EN) and the C-terminal extein (EC)) into a new intact peptide through a process called protein splicing (Supplementary Fig. 1)[5,6]. Inteins are usually small or can be reduced to a small domain close to 100 amino acids. Their function does not require any cofactors or energy source and they can typically work across relatively broad environmental conditions. Interestingly, an intein can be split into two parts, either naturally or artificially, without compromising its activity and thereby allowing protein trans-splicing[7], making such split inteins attractive tools in biotechnological fields[8]. The same features also allowed us to set up the SIMPL system.

Here, we describe the design and implementation of SIMPL, a live cell split-intein-based method for PPI detection that enables in situ analysis of interactions occurring in various cellular compartments as well as their responses to pharmacological challenges such as enzymatic and PPI inhibitors.

## Results

**Design and facilitation of SIMPL.** In our design (Fig. 1a), a bait protein is fused at its C-terminus to a V5 tag and an intein N-terminal fragment (IN). Correspondingly, a prey protein is fused at its N-terminus with a FLAG tag and an intein C-terminal fragment (IC). The bait and prey are co-expressed in selected mammalian cells to investigate their in vivo interaction. The association of bait and prey brings IN and IC into close proximity, allowing them to reconstitute a fully functional intein, which then catalyzes its own excision and the concurrent ligation of the bait and the prey peptides (as well as the V5 and FLAG tags). The resulting spliced protein can be resolved by regular western blot analysis due to its altered mobility, while the presence of the V5 and FLAG tags allows visualization or purification of protein using regular biochemical techniques.

The GP41-1 split intein, which was identified from environmental metagenomic sequence data[9], was chosen for use in the SIMPL system due to its small size (88 amino acids long in IN and 37 amino acids long in IC) and because it possesses the most rapid reaction rate among all split inteins examined[7,10,11].

Rapamycin-induced heterodimerization of FKBP1A (IC fused) and the FKBP rapamycin-binding (FRB) domain of mTOR[12] (IN fused) was used as a test case to evaluate SIMPL performance in a HEK 293 mammalian cell background. The major obstacle to implementing SIMPL is the intrinsic affinity between IN and IC, which introduces splicing unrelated to bait/prey interaction. We therefore re-engineered the GP41-1 split-intein. GP41-1 was re-split at eight different sites (Fig. 1b) and their behaviors were assessed (Fig. 1c). The intein split at position C25 (numbered from the last C-terminal amino acid of IC, Supplementary Fig. 2a) exhibited the best performance, with no apparent loss of enzyme activity and minimal self-association that is barely detected by western blot. The splicing reaction of C25 occurred with high fidelity, as only parental and spliced proteins are detected (Fig. 1c). This suggests that no N- or C-terminal cleavage occurred, which is a common side reaction of many split inteins[6,13]. The identity of the spliced protein was further verified by immunoprecipitation, where the proteins were pulled down by α-FLAG antibody, stringently washed, and probed with α-V5 antibody (or vice versa). In both cases only the spliced protein was detected and no obvious signal was observed in the sample without rapamycin treatment (Supplementary Fig. 2b). The C25 GP41-1 split intein was therefore adopted for use in our SIMPL system. It should be noted that the expression of FRB fused to WT IN, FRB-IN (C37), was hardly detected by western blot analysis, possibly as a consequence of fast degradation due to its severely disordered conformation. Moreover, extra bands appeared in the WT (C37) sample, indicating side cleavage products. Both deleterious effects were significantly reduced or abolished with all re-split inteins, suggesting a performance improvement achieved through resplitting.

To characterize the SIMPL system, we treated HEK 293 cells transiently transfected with FRB/FKBP1A SIMPL constructs with different concentrations of rapamycin (Fig. 1d). The results showed a typical dose–response relationship with a dose range similar to those measured by BRET-based methods[14]. A time course rapamycin treatment experiment also demonstrated a fast response, with interaction observed in as little as 2 min (the smallest observation interval used) and persistently accumulating over time (Fig. 1e). Similar kinetics were also observed in HeLa cells (Supplementary Fig. 2c) and PC9 lung adenocarcinoma cells (Supplementary Fig. 2d), suggesting that SIMPL can be applied to different mammalian cell lines. It should be noted that this time series signal profile is distinct from that observed with other methods: experiments performed using NanoBRET observed a rapidly reached equilibrium between association and dissociation[14]. This is derived from the differences in what various methods measure. Methods such as NanoBRET usually detect PPI complexes themselves. In contrast, SIMPL solely measures the event of protein association but not its dissociation or the steady-state complex.

We further assessed SIMPL in isogenic stable cells to examine potential problems derived from transient transfection such as uneven expression in various cells and difficulty in manipulating expression level. The stable cell line was created by incorporating both *FRB-IN* and *IC-FKBP1A* into the genome of host Flp-In T-Rex HEK 293 cells through Flp recombinase-mediated integration. Cells with different expression levels of both FRB and FKBP1A, induced by incubation with varied doses of tetracycline, were treated with rapamycin (Fig. 1f). Interaction-induced splicing and dose–responsive expression of FRB and FKBP1A was observed in all samples, even at the lowest dose of tetracycline (30 ng/ml) employed. Importantly, increasing protein expression was not accompanied by a significant increase in background signal as judged in samples without rapamycin treatment. These observations demonstrate that the SIMPL

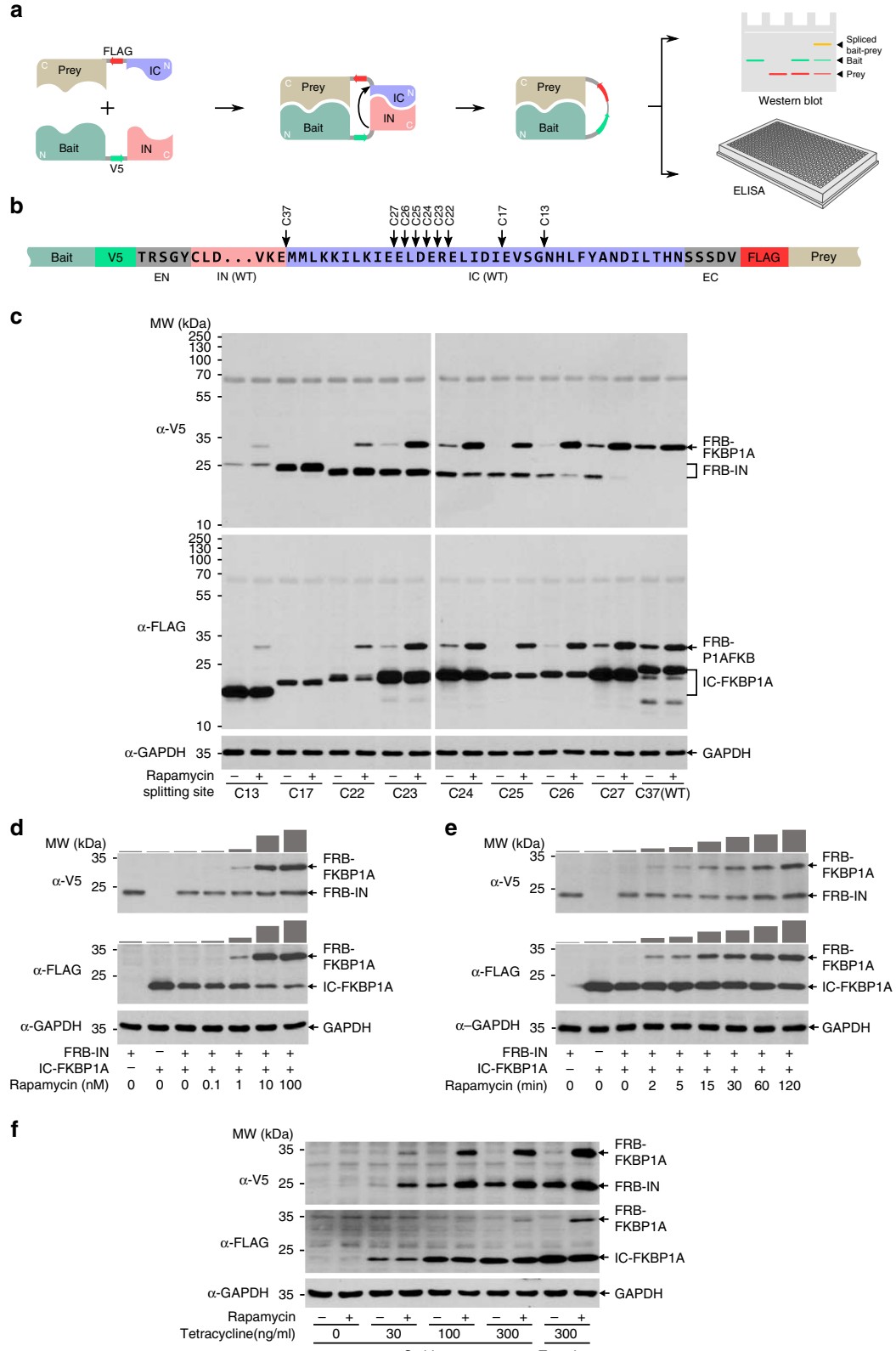

reaction is efficient, specific, and highly sensitive, and can work properly across a wide range of bait and prey expression levels.

**Alternative SIMPL formats to extend its detection capability.** In the above prototypic SIMPL design, a bait molecule is fused at its C-terminus to the IN fragment (IN format, or bait-V5-IN) and

a prey is fused at its N-terminus to the IC fragment (IC format, or IC-FLAG-prey). While functional in many cases, this arrangement limits the overall detection capability of SIMPL because in some instances the two tags in this format may be spatially inaccessible to each other. Additionally, the function of some proteins may be disrupted by the presence of tags on specific termini, necessitating a different strategy. We thus designed two

**Fig. 1 Development of the SIMPL assay. a** The design of SIMPL for PPI detection. **b** Schematic representation of SIMPL bait and prey constructs and the resplitting of the GP41-1 intein. **c** Examination of SIMPL system with GP41-1 split intein with different splitting sites. DNA constructs coding for FRB-IN and IC-FKBP1A with inteins split at the sites were expressed in HEK 293 cells. After incubation with rapamycin (100 nM) for 2 h, the cells were lysed and the lysates subjected to western blot analysis with α-V5 and α-FLAG antibodies. Both IN and IC constructs with C25 splitting exhibited the best performance and were adopted as the standard sensor intein for the SIMPL platform. The blot is representative of four independent experiments. **d** Dose response of rapamycin-induced FRB/FKBP1A interaction examined with SIMPL. HEK 293 cells expressing FRB-IN and IC-FKBP1A were treated with the indicated doses of rapamycin for 2 h followed by western blot analysis. The densities of spliced bands (FRB-FKBP1A) were quantified with ImageJ and are presented as bar graphs above the blots. The blot is representative of three independent experiments. **e** Time course of rapamycin-induced FRB/FKBP interaction. HEK 293 cells expressing FRB-IN and IC-FKBP1A were treated with rapamycin (100 nM) for different periods of time as indicated followed by western blot analysis. The densities of spliced bands (FRB-FKBP1A) were quantified with ImageJ and presented as bar graphs above the blots. The blot is representative of five independent experiments. **f** Stable cells derived from HEK 293 T-Rex FlpIn with *FRB-IN* and *IC-FKBP1A* inserted into the FRT site were treated with the indicated different concentrations of tetracycline for 16 h, followed by treatment with rapamycin (100 nM) for 2 h and then analysis by western blot. HEK 293 cells transiently transfected with FRB-IN and IC-FKBP1A were used as a control (right two lanes). The blot is representative of three independent experiments. Source data are available in the Source Data file.

alternative intein construct arrangements (Fig. 2a). In the C-terminal IC (CIC or prey-IC-FLAG) format IC is fused to the C-terminus of a prey while keeping the FLAG tag downstream. Its interaction with an IN bait leads to splicing between the bait (as well as V5 tag) and the FLAG tag, which produces a bait-V5-FLAG peptide. Similarly, the NIN (V5-IN-bait) format tags a bait N-terminally with the IN fragment and an upstream V5 peptide. Its interaction with an IC prey produces a V5-FLAG-prey peptide. Since both approaches lead to tag transfer, they still provide a readout of interaction that is compatible with western blot or IP-coupled western blot analysis. However, the interaction between an NIN bait and a CIC prey will produce a spliced small peptide V5-FLAG beyond the detection of western blot analysis. To address this, we created a CIC-GFP construct (prey-IC-FLAG-GFP) which can react with NIN bait to produce V5-FLAG-GFP peptide that allows detection by western blot or other analyses. Assessment of all four combinations of different SIMPL arrangements using the FRB/FKBP1A pair confirmed their feasibility, with rapamycin-induced bands corresponding to the molecular weight of appropriately spliced protein detected via western blot analysis in all cases (Fig. 2b, Supplementary Table 1). It should be noted that a basal splicing signal appeared in the sample of NIN/CIC-GFP combination without rapamycin treatment. This might be derived from an affinity change caused by different tagging or high-level expression of the proteins. However, the corresponding rapamycin-treated sample shows a dramatically increased signal (more than eight fold by density), making the states easily distinguishable from one other when proper controls and quantification are used. Thus, all four combinations are suitable for use in PPI detection in SIMPL.

**Enzyme-linked immunosorbent assay platform of SIMPL assay and its unbiased evaluation.** While use of SIMPL with a western blot readout is applicable to detailed PPI analysis, it is limited to low-throughput analyses and is not strongly quantifiable. We therefore developed an alternative, enzyme-linked immunosorbent assay (ELISA)-coupled SIMPL platform for high-throughput, quantifiable measurement of PPIs. For this purpose, a hemagglutinin (HA) tag was introduced into the bait construct in tandem with V5. This allows for monitoring of protein splicing using an ELISA format, with protein capture performed using α-FLAG antibody and detection performed using α-HA antibody coupled to horseradish peroxidase (HRP) (Fig. 3a). The SIMPL signal can be normalized to bait expression, which can be similarly measured by ELISA using immobilization with α-V5 antibody followed by detection with HRP-conjugated α-HA antibody (Fig. 3b). Performance of the ELISA platform was tested by monitoring the dose–response of rapamycin-induced FRB/FKBP1A interaction in all four combinations: IN/IC, IN/CIC,

NIN/IC, and NIN/CIC-GFP (Fig. 3c). The results of all four combinations showed an expected dose–response relationship consistent with the western analysis (Fig. 2b), demonstrating the feasibility of SIMPL ELISA. Specifically, the IN/IC profiles obtained from both ELISA analysis and quantified western analysis (Fig. 1d) presented notable similarity. Although two combinations, NIN/IC and NIN/CIC-GFP, showed relatively elevated basal splicing levels, these background signals were not high enough to interfere with interpretation of the spliced signal induced by rapamycin, with a robust 1.5- and 4-fold increase observed in the NIN/IC and NIN/CIC-GFP formats in ELISA analysis, respectively, after rapamycin treatment. In addition, expression of bait alone, either FRB (IN) or FRB (NIN), did not show any response to rapamycin treatment, further demonstrating the specificity of the assay.

We next evaluated the SIMPL system using a benchmarking approach with unbiased PPI reference sets, which has been widely accepted for assessing the overall performance of a PPI method. We employed a positive reference set (PRS) which contained 88 available positive PPIs derived from the previously well-established human PRS (hPRS)[15], including different types of PPIs and covering those occurring in various subcellular locations (Supplementary Table 2). Our random reference set (RRS) contained 88 protein pairs with baits and preys selected from the PRS but used in combinations determined computationally to have low probability of interaction (Supplementary Table 2). The reference sets were evaluated with the ELISA platform in two formats, bait(IN)/prey(IC) and bait(IN)/prey(CIC) (Supplementary Table 3). The expression of baits in the same samples was also tested with ELISA (Supplementary Fig. 3a) and each SIMPL signal was thereby normalized to its corresponding bait expression (Supplementary Fig. 3b). Receiver operating characteristic (ROC) analyses of the assays demonstrated exceptional sensitivity with AUC values of 0.806 and 0.867 for IN/IC and IN/CIC formats, respectively (Fig. 3d). Accordingly, 41% (IN/IC) or 56% (IN/CIC) of PPIs can be detected by SIMPL without compromising assay specificity and maintaining a false-positive rate of ~5% (Fig. 3e), determined with threshold values obtained from ROC analyses. We further compared the ELISA readout with western blot analysis by selecting 10 PPIs with high ELISA signals (above threshold) and 10 with low ELISA signals (below threshold) either in IN/IC or in IN/CIC format and re-testing them by western (Supplementary Fig. 3c, d). The results were highly consistent between the two methods; nine PPIs in the high IC group and all PPIs in high CIC group presented clearly observable splicing bands. In contrast, only one PPI in each low signal group presented a strong splicing signal relative to the levels observed in the high group bands. Comparison with the results from other PPI methods in the literature[16–18] shows

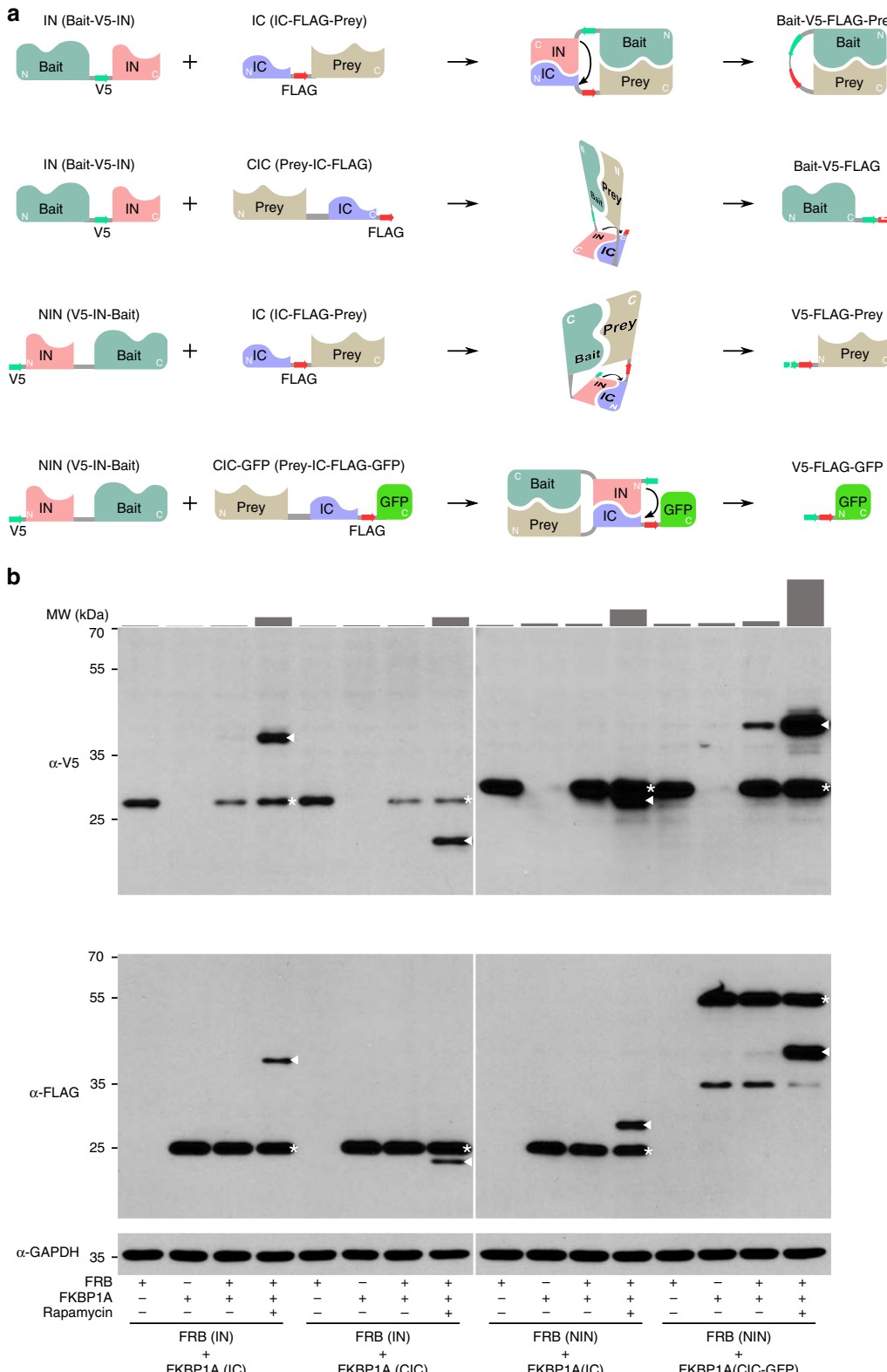

improvement of detection demonstrated with broader coverage (Fig. 3f). Interestingly, many PPIs can be detected by SIMPL using both the IN/IC and IN/CIC formats. Nevertheless, there are differences between the two orientations, presumably due to the spatial geometry of the interacting molecules or due to the disruption of the functionalities of the tagged termini. In general, the CIC format exhibits better performance in the tests described here (Fig. 3d, Supplementary Fig. 3e). For example, prey proteins

containing signal peptides showed better detectability in CIC format (Supplementary Fig. 3e) since tagging on N termini blocks their recognition by signal recognition particle and thereby affects their correct sorting to the membrane.

**Further characterization of SIMPL with physiological PPIs.** We next used SIMPL to explore physiological PPIs and chose PPIs in

**Fig. 2 Design of alternative formats of SIMPL to expand its capability. a** The IN/IC formats allow splicing between bait and prey. In the CIC orientation, IC-FLAG is fused to the C-terminus of a prey protein (Prey-IC-FLAG). Its combination with IN bait (Bait-V5-IN) leads to the transfer of FLAG tag to the bait generating Bait-V5-FLAG. In the NIN orientation, V5-IN is fused to the N-terminus of a bait (V5-IN-Bait). Its use with IC prey (IC-FLAG-Prey) causes the transfer of V5 tag to the prey thus generating V5-FLAG-Prey. The CIC-GFP construct (Prey-IC-FLAG-GFP) is created to allow the detection of NIN/CIC-GFP combination, which produces a V5-FLAG-GFP peptide. **b** The performance of different SIMPL formats was experimentally assessed using the rapamycin-induced FRB/FKBP1A interaction in which the corresponding bait and prey constructs were transiently transfected. Bands of spliced products are highlighted with triangles and parental proteins are highlighted with asterisks. The densities of spliced bands (FRB-FKBP1A) were quantified with ImageJ and are presented as bar graphs above the blots. The blot is representative of three independent experiments. Source data are available in the Source Data file.

the EGFR–RAS–ERK1/2 axis, an important signaling pathway[19]. Activated EGFR undergoes autophosphorylation and recruits scaffold proteins such as SHC1 to relay signal to downstream machinery[20]. At the RAS level, activated RAS (KRAS in this study) binds and directly activates RAF kinases[21] (RAF1 in this study). When EGFR-IN and IC-SHC1 were co-expressed in HEK 293 cells, their interaction was captured as a spliced band above EGFR recognizable by both α-FLAG and α-V5 antibodies (Fig. 4a). The interaction was also effectively detected using SHC1-CIC construct (Supplementary Fig. 4a). The interaction depends on EGFR activity as the constitutively active EGFR mutant (L858R) enhanced the signal while the kinase dead mutant (D855A)[22,23] abolished the interaction (Fig. 4a). To detect KRAS/RAF1 interaction, we chose the IC-KRAS construct to avoid the disturbance of its C-terminal lipidation. Assay with RAF1-IN detected the specific interaction (Fig. 4b) as wild-type KRAS and its constitutively active mutants (G12D and Q61H) displayed splicing signals, while no obvious signal was observed with the dominant negative KRAS mutant (S17N)[24]. Assay with the NIN-RAF1 construct exhibited a more marked splicing signal than RAF1-IN (more than sixfold by density, Supplementary Fig. 4b). It should be noted that the domain responsible for RAS binding is located at the N-terminus of RAF1 and hence the N-terminus is physically closer to RAS. We therefore speculate that the signal enhancement is caused by RAF1 N-terminal tagging, which improved accessibility of reactive termini. Thus, these two classical PPIs involved in normal and oncogenic signaling were successfully recapitulated by SIMPL.

As the study of rapamycin-induced FRB/FKBP1A interaction demonstrated the potential of SIMPL to follow interaction kinetics (Fig. 1e, Supplementary Fig. 2c, d), we tested whether SIMPL can also track kinetics of physiological PPIs using the example of EGF-activated SHC1 recruitment to EGFR. We found that interaction (splicing) between EGFR (IN) and SHC1 (IC) had already occurred without EGF treatment when they were transiently overexpressed in cells (Fig. 4a), and the signal was only slightly enhanced after EGF stimulation (Supplementary Fig. 4c) as the density ratio of the spliced bands between non-treated and EGF-stimulated (5 min) samples is 1:1.7. It should be noted that overexpression of EGFR induces ligand-independent autoactivation and subsequent recruitment of SHC1 (ref. [25]). Unlike phosphorylation and PPI, splicing is irreversible and therefore led to accumulation of the spliced protein. This accumulated spliced protein likely masked the signal induced by EGF stimulation. The key to overcoming this is to reduce the level and duration of bait and prey expression. Thus, we created a stable cell line derived from Flp-In T-REx HEK 293 cells by incorporating both *EGFR (IN)* and *SHC1 (IC)* into the genome through Flp recombinase-mediated integration. This allowed control of the amplitude and temporal induction of their expression by tetracycline. Notably, in cells treated for 6 h with tetracycline in starvation medium (0.1% fetal calf serum) the splicing signal of EGFR/SHC1 interaction was markedly reduced, but restored upon EGF stimulation for 2 min (Fig. 4c), supporting

the feasibility of using SIMPL to follow physiological PPIs. It should be noted that, due to the leakage of the tetracycline repressor system, low levels of EGFR (IN) and SHC1 (IC) were observed in cells without tetracycline treatment. EGF-stimulated splicing was also observed under this condition, consistent with the strong sensitivity of SIMPL assay.

As most PPI methods are not well suited for detection of weak or transient PPIs, we wanted to test whether SIMPL is capable of following these types of interactions. We selected protein kinases as our test case since their association with substrates, similar to common enzyme/substrate interactions, is usually characterized as transient and weak[26]. Thus, IN-fused MAPK1 (ERK2), MAPK8 (JNK1), MAPK14 (p38α MAPK), MAPK7 (ERK5), AKT1, and PRKCA (PKCα) were examined with several well documented substrates in both IC and CIC formats. ELISA assay demonstrated that more than 50% of these PPIs could be detected by SIMPL, in one or both formats (Fig. 4d). MAPK members presented relatively strong interactions, which may be enhanced by docking interaction outside of their active sites[27]. We examined whether SIMPL could capture the kinetic process of kinase/substrate interactions. Indeed, we observed an enhanced MAPK1/ELK1 interaction after MAPK1 activation induced by tetradecanoylphorbol acetate (TPA) stimulation (Supplementary Fig. 4d). However, no obvious increase was observed in many cases of kinase activation (Supplementary Fig. 4e–g). We speculate that in these instances the accumulation of the spliced proteins from the basal state may have masked the stimulation response due to the irreversibility of the splicing reaction and might be avoided by reducing basal bait/prey expression through the use of the stable cell line approach mentioned above for EGFR/SHC1, an area which deserves further investigation.

The evaluation with the reference PPI set suggests SIMPL is capable of detecting PPIs in various cellular compartments as the PRS covers PPIs occurring in different locations such as nucleus, cytoplasm, plasma membrane, and extracellular space (Supplementary Table 2). We further tested this by examining mitochondrial PPIs with SIMPL as mitochondria are special organelles with distinct features and their PPIs are often difficult to study. We selected several well-studied mitochondrial PPIs for this purpose (Supplementary Table 4), including proteins involved in oxidative phosphorylation[28], transport[29], cristea organization[30], and metabolism[31]. Baits were prepared using the IN format to avoid interference of transit peptides usually at N-termini of mitochondrial proteins. The corresponding preys were constructed in either the IC or CIC configuration (or both) to reduce the chance of steric interference preventing their association with IN or to prevent incorrect sorting of the prey proteins. Out of 10 PPIs examined, 8 (TIMM50/TIMM23, PDHA1/PDHB, CHCHD6/CHCHD3, NDUFV1/NDUFV3, SDHA/SDHB, UQCRC2/UQCRC2, ATP5MC1/ATP5MC1, and ETFA/ETFB) were successfully detected (Fig. 4e), including proteins localized to different sub-mitochondrial compartments (matrix, inner membrane and intermembrane space). We tested two additional controls for PDHA1/PDHB interaction to exclude

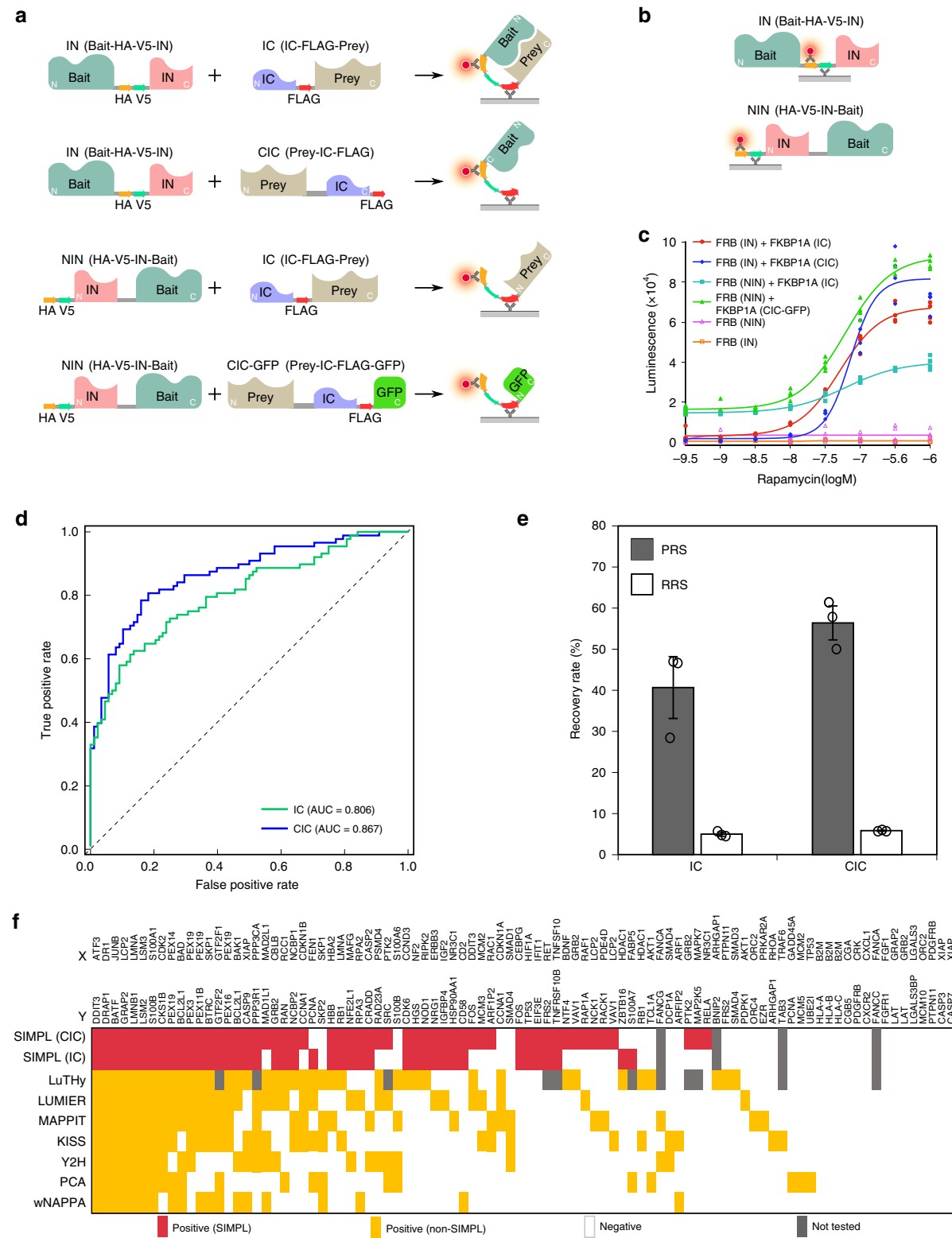

the possibility that splicing occurred in cell lysates during sample processing (Fig. 4f). In the first control, the mitochondrial targeting sequence of PDHA1 was replaced with the nuclear localization signal of MYC. This modification prevents the resultant NLS-PDHA1 from being sorted to the mitochondria and, consistent with this, its interaction with PDHB was therefore not observed. In the second control, PDHA1 and PDHB were

separately expressed in different cells and the two cell lysates were mixed and assayed by ELISA. Once again, no interaction between PDHA1 and PDHB was detected. Together, these results indicate that a significant amount of PDHA1/PDHB SIMPL splicing does not occur if both proteins are not properly targeted to the mitochondrion, supporting the broad applicability of SIMPL for monitoring interactions with diverse subcellular localizations.

**Fig. 3 Setting up the SIMPL ELISA platform and its evaluation with reference PPIs. a** An extra HA tag is introduced into the bait construct. The spliced proteins are captured by immobilized α-FLAG antibody and measured with α-HA antibody conjugated with HRP. All four SIMPL formats described in Fig. 2 are compatible with ELISA. **b** The bait proteins can be measured similarly by ELISA using immobilized α-V5 antibody and HRP-conjugated α-HA antibody probe. **c** The ELISA platform was assessed with the rapamycin-induced FRB/FKBP1A interaction. HEK 293 cells expressing FRB and FKBP1A in different formats were treated with different doses of rapamycin as indicated for 30 min followed by lysis and ELISA analysis. The experiment was performed with four technical replicates and each replicate is presented as a single dot. **d** Benchmarking analysis of the overall performance of SIMPL ELISA platform. Eighty-eight PPIs well documented in literature were chosen as positive reference set (PRS). Eighty-eight pairs of bait/prey combinations with the least possibility of interaction were selected from the bait and preys of the PRS to form the random reference set (RRS). Both sets were then screened using SIMPL ELISA analysis in both IN/IC and IN/CIC formats. The spliced signal was normalized to bait expression. Receiver operating characteristic (ROC) analysis was performed as presented. Data shown here are a representative result of three experiments. **e** Performance of the SIMPL assay in terms of sensitivity (true-positive rate) and false-positive rate (1-specificity). The threshold values for positive detection were determined from ROC analysis as in **d**. Results are averages of three independent experiments showing mean recovery rate ± SEM. **f** Comparison of SIMPL detection of individual PPIs in the PRS to results from seven different PPI methods obtained from the literature[16–18]. Source data are available in the Source Data file.

**Detecting PPIs in *C. elegans* with SIMPL**. We next investigated the feasibility of using the SIMPL system in a multicellular animal using the nematode *Caenorhabditis elegans* as a model. We selected a *C. elegans* PRS of 27 PPI pairs from previously identified literature confirmed interactions[32] and from interactions used previously to evaluate binary PPI mapping approaches[33] (Supplementary Table 5). We also assembled a *C. elegans* RRS by randomly combining protein pairs from the PRS, excluding known interactors. Full-length ORFs of the corresponding genes were PCR amplified and cloned into vectors containing split-intein tags optimized for expression in *C. elegans*, but otherwise identical to the ELISA compatible split-intein fragments used above.

Proteins were expressed under control of the general *rps-0* ribosomal promoter. Transgenic animals were generated by microinjection, and both IC and CIC configurations were injected for each prey protein. To enable accurate quantification of splicing by ELISA, we injected a 4× higher concentration of prey plasmid than bait plasmid to ensure that splicing of the bait protein is not limited by the availability of the prey protein. All transgenic lines were first tested for expression of both bait and prey protein by western blotting. In all, we recovered 10 PRS pair-expressing lines and 13 RRS pair-expressing lines, representing 7 and 9 unique protein pairs, respectively (Fig. 5a). We first analyzed each line for splicing by western blot (Supplementary Fig. 5). We observed visible splicing for 7/10 PRS pairs and 5/13 RRS pairs; however, levels of splicing for the RRS pairs were lower than for the PRS pairs. We then quantified levels of splicing using ELISA as above, analyzing two independent protein lysates for each transgenic line. Using a sliding cutoff level of spliced bait/total bait signal to assign positive interactions, we observed clear separation between PRS and RRS protein pairs in both replicates (Fig. 5b). At a cutoff level that optimizes the fraction of true positives vs. false positives detected in both replicates, 8/10 PRS pairs tested positive in both replicates, while the remaining 2 pairs tested positive in one replicate (Fig. 5c). In contrast, none of the RRS pairs tested positive in both replicates, though six pairs tested positive in a single replicate. Collapsing IC/CIC orientations, 7/7 PRS protein pairs tested positive, and 1/9 RRS pairs tested positive. Overall, these results indicate that the SIMPL system is functional in *C. elegans* and suggests its universal nature could allow it to be exploited in many other systems.

**SIMPL as a drug screening platform**. Finally, we investigated whether SIMPL can serve as a drug screening tool, in particular as an assay that could detect enzymatic- as well as PPI inhibitors. Since protein splicing is an irreversible process, inhibitors have to be administered before bait/prey expression (Fig. 6a). Using the EGFR/Shc1 interaction as an example, we observed a decrease of SIMPL signal upon the administration of AG1478 (ref. [34]), an

EGFR tyrosine kinase inhibitor (TKI) which suppresses EGFR autophosphorylation and thereby reduces EGFR/Shc1 interaction (Fig. 6b). In this case, the interaction between Shc1 and EGFR occurs downstream to EGFR activation and serves as an indirect readout of EGFR activity. We then tested whether SIMPL can also monitor PPI inhibition caused by a direct PPI inhibitor using venetoclax, an FDA approved drug targeting the BAX/BCL2 interaction[35], as an example. We first examined the interactions of BAX/BCL2 (and BAX/BCL2L1 control) in three SIMPL formats, IN/IC, IN/CIC, and NIN/IC (Fig. 6c–f). Note that for the NIN-BAX/IC-BCL2 combination, immunoprecipitation was performed to resolve spliced protein product from the similarly-sized parental product (Fig. 6e). Based on these results the NIN-IC pairing displayed the highest signal (Fig. 6f). Using this combination in SIMPL with an ELISA readout we observed potent inhibition of the BAX/BCL2 interaction in the presence of venetoclax with an $IC_{50}$ of 9.1 nM (very similar to other cell-based assay results[36]) but not the unrelated control TKI osimertinib (Fig. 6g, red and blue). The effect of venetoclax on BCL2L1 (BCL-XL) was much less potent as the inhibition of the BAX/BCL2L1 PPI was observed only at high venetoclax concentration (Fig. 6g, green), consistent with previous reports[36]. Overall, these results clearly demonstrate the potential for using SIMPL as a highly sensitive small-molecule screening tool.

## Discussion

There is no universal method that works for every PPI and all interaction proteomics methods exhibit preferences, leading to low overlap of PPI coverage across many approaches[3,37]. We developed SIMPL to expand detection coverage. PRS derived from hsPRS-v1 (ref. [15]) is used to benchmark the suitability of SIMPL and to unbiasedly compare it to seven different PPI assays: LuTHy[17], LUMIER[38], MAPPIT[39], KISS[16], Yeast Two-Hybrid (YTH)[40], PCA[41], and wNAPPA[42]. Our data demonstrate the improvement of PPI detection in terms of sensitivity, coverage, and the objective parameter AUC. We further tested SIMPL in *C. elegans* and obtained similar results. Therefore, SIMPL is a sensitive method for PPI detection with a low rate of false positives (≤5%) and suitability for use in different in vivo models.

Many features of the split-intein sensor make SIMPL a unique system. Since inteins do not exist in mammals it is unlikely that they would interfere with the function of mammalian cells. Indeed, we have not observed any cellular changes when inteins are orthogonally expressed in mammalian cells. The small size of GP41-1 split intein used in SIMPL also reduces its potential for involvement in nonspecific interactions. Additionally, since GP41-1 can work under relatively broad physiological conditions, it allows detection in different subcellular locations. Unlike many PPI methods, extra cofactors are also not needed in the SIMPL assay.

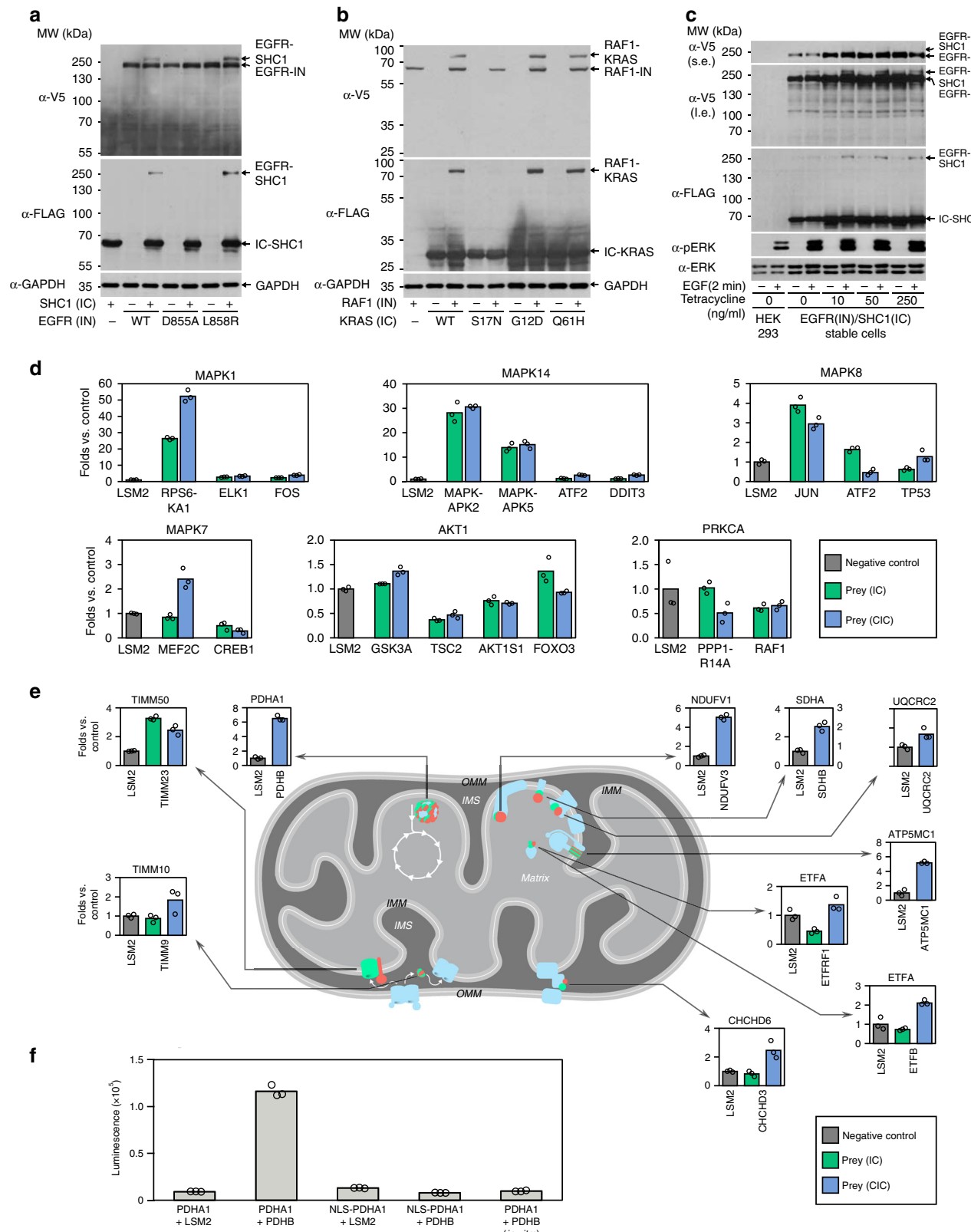

The kinetic features of split inteins should be considered. It has been shown that separated split intein is partly folded or completely disordered. The recognition between IN and IC initiates the association process, causes a disorder-to-order transition, and then triggers splicing[43,44]. The C25 re-split GP41-1 developed in this study presumably proceeds through a similar kinetic route except that the association is mainly driven by bait/prey interaction, not by the recognition between IN and IC, due to the shortening of the IC fragment. Indeed, most PCA-based PPI sensor fragments are in unfolded conformations before complementation[45]. In contrast to GP41-1, most of them, including BiFC, follow slow complementation kinetics which

**Fig. 4 Detection of physiological PPIs and their inhibition with SIMPL. a** EGFR/SHC1 interaction. EGFR WT, inactive (D855A), or constitutively active (L858R) mutants in IN format were co-expressed with SHC1 (IC) in HEK 293 cells. Their interactions were analyzed with western analysis. **b** KRAS/RAF interaction. RAF1-IN and IC-KRAS (WT, inactive S17N mutant, or active G12D or Q61H mutant) were transiently expressed in HEK 293 cells followed by western analysis. **c** Stable cells derived from HEK 293 T-Rex FlpIn with *EGFR (IN)* and *SHC1 (IC)* inserted into the FRT site were treated with the indicated different concentrations of tetracycline for 6 h, followed by treatment with EGF (100 ng/ml) for 2 min and then analysis by western blot. s.e. short exposure, l.e. long exposure. Each blot in **a**–**c** is representative of three independent experiments. **d** SIMPL analysis of kinase/substrate interactions. The indicated kinase-IN constructs were individually expressed along with their substrates in either IC or CIC format and their interactions were detected by ELISA assay. LSM2 was used as a negative control prey. **e** Mitochondrial PPIs. The selected mitochondrial bait proteins were constructed in IN format. They were then co-expressed with the indicated preys in either IC or CIC format, or in both. The interactions were examined with ELISA. LSM2 was used as a negative control prey. OMM outer mitochondrial membrane, IMS intermembrane space, IMM inner mitochondrial membrane. **f** Retest of PDHA1/PDHB interaction with ELISA-coupled SIMPL. In one sample (the fourth from left), the mitochondrial targeting sequence of PDHA1 was replaced by a nuclear localization sequence. In the last sample (the first from right), WT PDHA1 and PDHB were expressed in separate cells and the cell lysates were mixed for ELISA. The experiments in **d**–**f** were performed in triplicates and their mean values are presented as bars with each replicate shown in a single dot. Source data are available in the Source Data file.

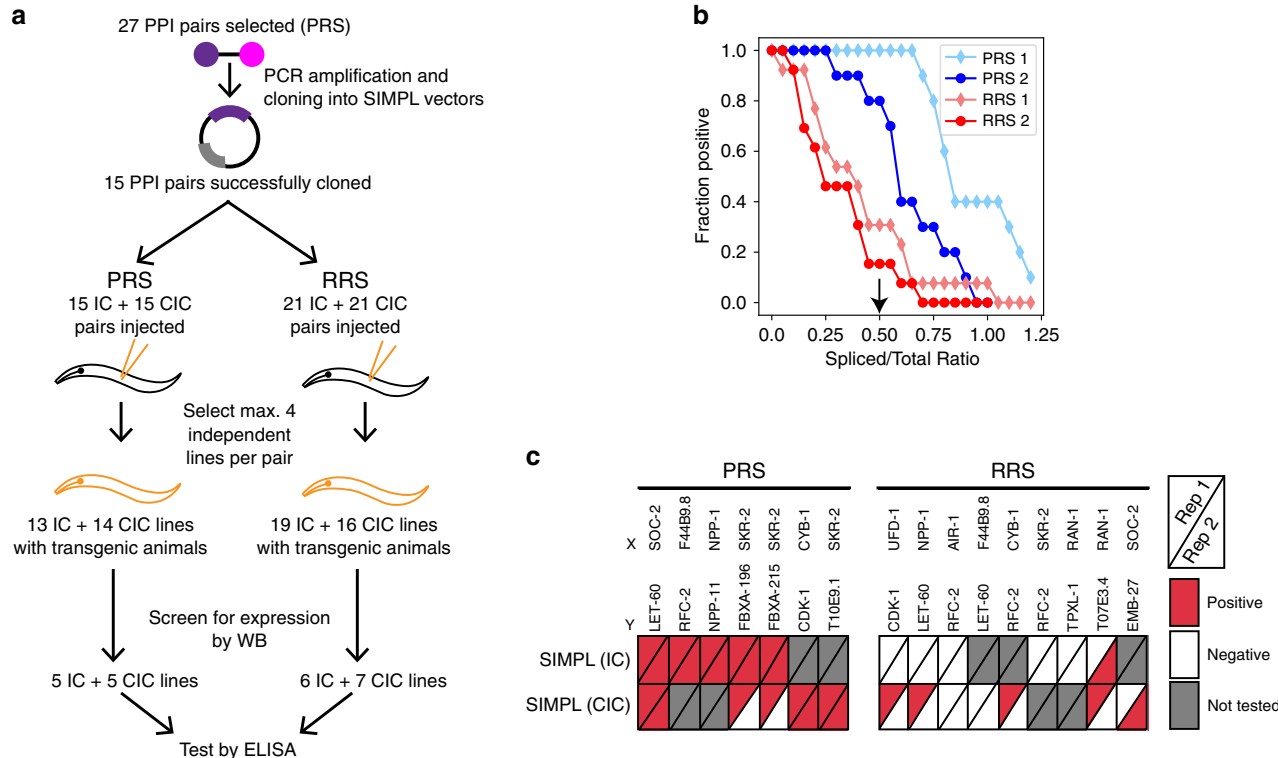

**Fig. 5 Detecting PPIs in *C. elegans* with SIMPL. a** Scheme for creating and testing transgenic *C. elegans* lines for use with the SIMPL ELISA assay. Initially, 15 positive reference set (PRS) PPI pairs were successfully cloned and injected into *C. elegans*, along with 21 random (RRS) PPI pairs, in both the IN/IC and IN/CIC configurations (72 total lines injected). Potential transgenic lines were screened by western blot for expression of both bait and prey constructs, resulting in 10 total PRS lines and 13 total RRS lines (IC + CIC), representing 7 and 9 distinct PPI pairs, respectively. **b** Performance of the SIMPL ELISA assay in two biological replicates, with the fraction of PPI pairs that are positive at different cutoff values. Cutoffs were set as the ratio of spliced signal to total signal. A biological replicate was reported as the average value of two technical replicates. **c** Visualization of SIMPL ELISA results of the individually tested PPIs. The cutoff value used to determine whether a given PPI was positive or negative was 0.50 (arrow in **b**), which maximized the number of true-positive and true-negative interactions between both biological replicates. Pairs belonging to the RRS were never positive in both replicates compared to the PRS. Source data are available in the Source Data file.

impairs their ability to detect fast interactions. The physical mechanism of this ultrafast kinetics of GP41-1 merits full investigation. It should be noted that luciferase-based PCA and some other PPI methods such as FRET, BRET[3], and the recently developed SPARK (Specific Protein Association tool giving transcriptional Readout with rapid Kinetics)[46] are also capable of tracking fast PPIs, and, alongside SIMPL, provide a robust set of complementary tools for studying these PPIs and their kinetics in vivo.

The most striking feature of SIMPL is its irreversibility. Complementation of many PCA sensors is also irreversible but due to energetic trap. The covalent splicing in SIMPL makes it useful even with harsh experimental conditions, helping avoid the loss of specific interactions (and the gain of nonspecific interactions) during processing steps, which are common problems for many affinity-based methods. It should be noted that irreversible sensors actually record the association event of complex formation, not the complex itself. The accumulation of "unbreakable" signal may also substantially enhance the sensitivity. This feature, together with the ultrafast activity of GP41-1, makes SIMPL well suited to following PPI kinetics. However, kinetic studies with SIMPL need to be carefully designed since the accumulation of

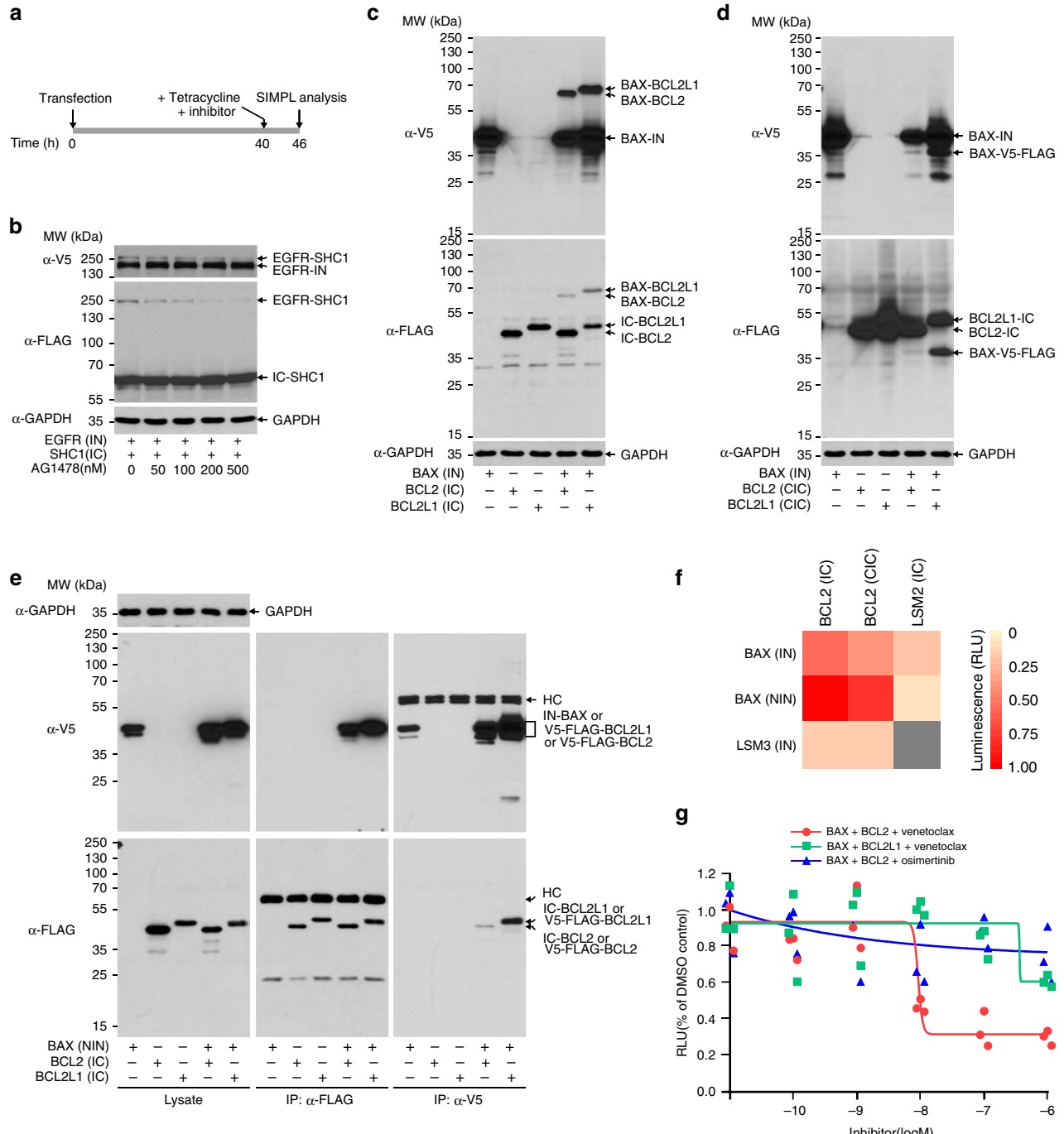

**Fig. 6 SIMPL for enzymatic/PPI inhibitor identification. a** Time schedule for studying enzymatic/PPI inhibitors with SIMPL. To avoid splicing before inhibition can occur, an inhibitor has to be administered before protein expression. As the expression is under the control of Tet-on promoter, tetracycline is added to the cells alongside the inhibitor 6 h before assay. **b** Studying an EGFR kinase inhibitor with SIMPL. EGFR kinase inhibitor AG1478 at different indicated doses was incubated with the cells expressing EGFR-IN and IC-SHC1. The spliced EGFR-SHC1 band observed by western blot diminished with increasing AG1478 concentration. The blot is representative of three independent experiments. **c–e** BAX/BCL2 interaction was assayed in different formats as indicated by western analysis. In the case of NIN-BAX/IC-BCL2, immunoprecipitation was performed to resolve the spliced protein from its parental protein since they have similar mobility upon electrophoresis (**e**). Each blot is representative of three independent experiments. **f** Heatmap of SIMPL ELISA readings of BAX/BCL2 interaction in different formats. Gray color: not tested. LSM3 and LSM2 were used as negative controls for bait and prey respectively. **g** Investigation of the BCL2/BAX PPI inhibitor venetoclax with SIMPL. Cells expressing NIN-BAX and IC-BCL2 (or BCL2L1 as control) were treated with venetoclax (or osmeritinib as a negative control) at different concentrations as indicated. The cells were then subjected to ELISA analysis. SIMPL signals were normalized to bait expression. The experiment was performed in triplicates and each replicate is presented as a single dot. Source data are available in the Source Data file.

basal interaction signal can mask the signal of nascent complex formation if the basal PPI is relatively high. In this study, we used inducible expression of stably incorporated genes to help reduce basal interaction signal.

Our study also demonstrates the feasibility of SIMPL as a platform to characterize enzymatic and PPI inhibitors. Inhibitor efficiency measured with SIMPL was very close to results obtained with other methods, indicating that SIMPL can serve as

an attractive, live cell-based assay for drug screening. Additionally, SIMPL may help overcome the challenges associated with identifying compounds that specifically disrupt or enhance PPIs which, though promising candidates for therapeutic intervention, have been difficult to target to date.

Notably, split inteins have previously been used for PPI detection, though in a more complex format, wherein the two intein moieties are respectively fused to components of split reporters such as fluorescent protein or luciferase[8], and the readout is production of functional spliced reporter molecules. Importantly, the split inteins used in the aforementioned approach have a tendency to self-associate[7], a process that is further aggravated by the affinity between the halves of the split reporter molecule, potentially resulting in a significant increase in nonspecific PPI signal. For these reasons, the method has not been widely used in research. In contrast, we have developed a straightforward split-intein strategy which allows protein splicing between bait and prey, or small tags, minimizing introduction of extra components. Importantly, we have also re-engineered our split-intein sensor to overcome the natural affinity between the intein components. Both factors dramatically decrease any nonspecific interaction derived from the detection system.

As with any other PPI method, SIMPL does have certain limitations. As heterologous expression of bait and prey proteins is required, SIMPL cannot measure endogenous PPIs unless the tags are integrated into the target genes by gene editing. Additionally, SIMPL is not able to follow protein dissociation due to its irreversibility. Despite these limitations, however, the data presented here demonstrate several attractive assets of SIMPL: improved high sensitivity without loss of specificity, broad applicability to PPIs in various cellular compartments or in different cell types, the ability to detect weak and transient PPIs, the potential to follow PPI kinetics, quantifiability, compatibility with high-throughput screening, and no requirement for special equipment or expertise. We therefore envision that SIMPL will have broad applications in biomedical research as well as the pharmaceutical industry.

## Methods

**Molecular cloning and library preparation**. The plasmids containing GP41-1 split-intein cDNA, pCAG-Co-InCreN and pCAG-Co-InCreC, were obtained from Addgene. SIMPL bait and prey vectors are generated by integrating DNA pieces of GP41-1 split-intein fragments, linkers, and tags, as well as Gateway cloning cassette, into pCMV5 vector backbone by Gibson assembly (New England BioLabs). Plasmids for FlpIn stable cloning were created similarly into pCDNA5/FRT/TO vector with both bait and prey included by Gibson assembly. Most cDNAs were originally obtained from human ORFeome collection or from the Openfreezer collection at Lunenfeld-Tanenbaum Research Institute[47]. Those not in entry clone vectors were cloned into pDONR223 by PCR and Gateway BP reactions (Life Technologies). Different cDNA fragments were then cloned into SIMPL vectors by Gateway LR reactions (Life Technologies). Site-directed mutagenesis was generated by PCR using KAPA HiFi DNA polymerase (KAPA Biosystems). The plasmids created in this study are available from the corresponding author upon reasonable request.

**Cell culture and treatment**. HEK 293, HEK 293 Flp-In T-Rex, and HeLa cells were grown in DMEM supplemented with 10% fetal calf serum (Life Technologies). PC9 cells were grown in RPMI 1640 medium supplemented with 10% fetal calf serum. For ELISA assay, cells were seeded in 96-well (Sarstedt AG & Co) or 384-well plates (Greiner Bio-One) with 15,000 (96 well) or 5000 (384 well) cells per well. Cells were transfected with various plasmids with polyethylenimine (PEI) Max (Polysciences)[48]. Expression in HEK 293 Flp-In T-Rex cells was induced by treating the cells with tetracycline (1 μg/ml) for 6–16 h. For experiments to study signaling pathway activation, the cells were starved with 0.1% fetal calf serum as well as treated with tetracycline for 6 h before stimulation with EGF (Sigma-Aldrich), tetradecanoylphorbol acetate (TPA) (Sigma-Aldrich), or anisomycin (Sigma-Aldrich). Stable cell lines were created according to the manual of Flp-In T-REx (Invitrogen). Briefly, plasmid containing both bait and prey DNA and pOG44 plasmid (1:10 ratio) were cotransfected into HEK 293 Flp-In T-Rex cells. After 3 days, the cells underwent puromycin selection. Single colonies were selected and the expression of bait and prey were verified by western blot analysis.

**Western blot analysis and immunoprecipitation**. Cells were lysed in buffer H (Triton X-100 1%, β-glycerophosphate pH 7.3 50 mM, EGTA 1.5 mM, EDTA 1 mM, orthovanadate 0.1 mM, DTT 1 mM supplemented with protease inhibitors (Roche)). After centrifugation at 15,000 r.p.m. for 10 min, the supernatants were mixed with Laemmli sample buffer, boiled at 95 °C for 3–5 min, and subjected to western blot analysis. For immunoprecipitations, supernatants (0.3 ml) were incubated with antibodies at 4 °C with rotation for 1 h, followed by another hour of incubation with protein G sepharose beads (GE Healthcare). Beads were washed twice with LiCl (0.5 M in Tris pH 8.0 0.1 mM) and twice with lysis buffer, boiled with Laemmli sample buffer, and then were subjected to western blot analysis. Antibodies used for western blot analysis and immunoprecipitations were: α-FLAG antibody purchased from Sigma-Aldrich Co. (F1804) with 1:10,000 dilution and α-V5 antibody from Cell Signaling Technology (#13202) with 1:10,000 dilution. Each of the above antibodies was diluted according to provider's protocol.

**ELISA**. HEK 293 cells were grown in 96- well or 384-well plates and were transfected with PEI as aforementioned. The cells in each well were lysed in 120 μl (96 well) or 80 μl (384 well) TNE buffer (Tris pH 7.5 20 mM, NaCl 150 mM, EDTA 2 mM and Triton X-100 0.5% supplemented with protease inhibitors). Aliquots of lysates (20 μl) were incubated for 3 h at 4 °C in a well of a 384 well Lumitrac plate (Greiner Bio-One) that was coated with α-FLAG antibody (20 μl/well with 1:100 dilution) and blocked with BSA. After three times thorough wash with phosphate buffer saline supplemented with 0.05% Tween 20 (PBST), the plate was incubated with HRP-conjugated α-HA antibody (GeneTex GTX115044, 1:5000 dilution) for 1 h at room temperature. The plate was washed three times with PBST followed by chemiluminescence reading using SuperSignal ELISA Pico substrate (ThermoFisher).

**Selection of RRS pairs**. All bait-prey pairs (75 baits × 78 preys) were considered for the RRS, and 88 were selected that had the lowest chances of interaction, using the following criteria: (1) absence from the PRS; (2) absence from the Integrated Interactions Database ver. 2018-11 (ref. [49]), thereby ensuring that the pairs had not been detected in experimental studies, predicted based on orthology, or predicted by five computational algorithms; (3) lowest probabilities of interaction according to the FpClass PPI prediction algorithm[50]; and (4) maximal coverage of candidate baits and preys.

**C. elegans SIMPL vectors and cloning**. To facilitate assembly of the expression plasmids, we used a SapI-based cloning strategy. We generated a series of donor vectors based on the kanamycin-resistant cloning vector pHSG298 (Takara Bio), in which the insert is flanked with SapI-sites. Digestion with SapI yields overhangs that enable assembly of promoter, ORF, split-intein, and UTR into a destination vector (pMLS257 Addgene #73716) in a single ligation reaction. The following plasmids were generated: (i) donor plasmids containing IN (pJRK244), IC (pJRK036), and CIC (pJRK152) split-intein donor sequences. Split-intein amino-acid sequences are identical to the mammalian ELISA compatible split-intein constructs, but are codon optimized for C. elegans and contain an artificial intron. (ii) Two rps-0 promoter donor plasmids, pJRK001 for assembly with IC and IN, and pJRK151 for assembly with CIC. (iii) Three unc-54 3′-UTR plasmids: pJRK150 for assembly with IC, pJRK153 for assembly with CIC, and pJRK002 for assembly with IN. ORFs were amplified by PCR from a mixed-stage cDNA library and cloned blunt-ended into vector pHSG298 digested with Eco53kI. Plasmid sequences available upon request. Plasmids used for injection were purified using the PureLink HQ Mini Plasmid DNA Purification Kit (ThermoFisher) using the extra wash step and buffer recommended for endA+ strains.

**C. elegans strain and culture conditions**. C. elegans strains were cultured under standard conditions[51]. Only hermaphrodites were used and all experiments were performed with animals grown at 20 °C on nematode growth medium agar plates seeded with E. coli OP50 bacteria.

**Extrachromosomal strain generation**. Young adult N2 animals were injected with 20 ng/μl of the prey IC/CIC SIMPL plasmid, 5 ng/μl of the bait IN SIMPL plasmid, 20 ng/μl of a plasmid conferring a dominant Rol phenotype and Hygromycin B resistance (pDD382 Addgene #91830), and 55 ng/μl lambda DNA (Thermo-Scientific SM0191). Four hermaphrodites were injected for each protein pair and placed on individual plates. After 2–3 days, Hygromycin B (250 μg/ml) was added to the plates to select for transgenic lines. From each plate a single F2 Rol animal was picked to establish up to four transgenic strains per protein pair, and each was tested for expression of the SIMPL constructs.

**C. elegans lysis and ELISA**. Mixed-stage animals grown under Hygromycin B (250 μg/ml) selection were washed off with M9 buffer (0.22 M KH$_2$PO$_4$, 0.42 M Na$_2$HPO$_4$, 0.85 M NaCl, 0.001 M MgSO$_4$), and washed two more times with M9 buffer. Samples were then pelleted and resuspended in 100–400 μl of Lysis Buffer (25 mM Tris HCl pH 7.5, 150 mM NaCl, 1 mM EDTA, 0.5% Igepal 630, and 1 tablet/50 ml complete protease inhibitor cocktail (Sigma-Aldrich)). After flash freezing in liquid nitrogen and thawing, samples were sonicated with a Diagenode

BioRupter Plus, 5 min high setting: 30 s on/30 s off in a 4 °C water bath. The lysates were then spun at max speed in a tabletop centrifuge at 4 °C for 15 min to clear cellular debris. ELISA was performed as above, but the SuperSignal ELISA pico chemiluminescent substrate was used undiluted (ThermoScientific).

**Statistics**. Two-tailed Student's $t$-test with $n = 3$ was used to examine the significance of kinase/substrate and mitochondrial PPIs.

**Reporting summary**. Further information on research design is available in the Nature Research Reporting Summary linked to this article.

## Data availability

All data generated or analyzed during this study are included in this published article (and its Supplementary Information files). Source data for the figures presented in the main manuscript and the Supplementary Information are available in the Source Data file. All other relevant data are available from the authors upon reasonable request.

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

## Acknowledgements

We thank Punit Saraon for the help with the figure preparation, Jelena Tomic for editing, Mandy Lam and Mikko Taipale for providing access to equipment, and Mandy Lam, Mikko Taipale and Frederick Roth for providing cDNA clones, R. Schmidt for help with *C. elegans* strain generation. The work in the Stagljar lab is supported by the Cancer Research Society (#505486), Genome Canada via Ontario Genomics (#OGI-117, #OGI-159 & #OGI-120), Ontario Research fund (ORF/DIG-501411 & RE08-009), CIHR (#PJT-165862), FACIT Prospects Oncology Grant (#507942), and CQDM (Quantum Leap). This work was also supported by the Netherlands Organization for Scientific Research (NWO) VICI grant 016.VICI.170.165 to M.B. Jurisica lab was in part supported by Ontario Research Fund (#34876), Natural Sciences Research Council (NSERC #203475), and Canada Foundation for Innovation (CFI #225404, #30865).

## Author contributions

Z.Y. conceptualized SIMPL, was actively involved in most experiments and data analysis, and wrote the bulk of the manuscript. F.A. performed some experiments. F.A., I.A., and P.T. were involved in DNA preparation. J.S. contributed to manuscript writing. J.S. and A.L. helped creating stable cell lines. J.K. and M.B. undertook the *C. elegans* work and wrote the related part in the manuscript. I.J. and M.K. performed computational analysis of RRS, helped data analysis, and contributed to manuscript writing. I.S. guided and supervised the work, contributed to manuscript writing and coordinated the preparation of the manuscript. All authors approved its content.

## Competing interests

I.S. and Z.Y. are named as co-inventors on a patent application concerning the described technology.
