## [Peer Review File · Nature Communications]

Reviewers' Comments:

Reviewer #1:

Remarks to the Author:

In this paper, Yao and colleagues report the development of a method to detect protein-protein interactions (PPIs) based on the trans-splicing reaction mediated by the split intein gp41-1. The authors justify why they decided to develop such method saying that the current methods have several limitations, for instance the inability to detect weak and transient interactions or to function in all cellular compartments. While I do agree that good methods to robustly detect PPIs are very important, I fail to see how the method presented here represents the solution to the problem.

Weak and transient interactions can be detected to date either using complementation of fluorescence (BiFC) or transcriptional readouts. In both cases, transient interactions are converted in a long-lasting response (either fluorescence which, unless specifically engineered to be labile, is quite stable, or a gene product). Few years ago, a method called SPARK was developed that combines a transcriptional readout with light activation, so that even PPI dynamics can now be tracked. The authors do not cite the SPARK method (Kim et al., *elife*, 2017), however in my eyes this method is more powerful than the one described here. It is also compatible with high-throughput and can be used for drug screening.

BiFC-based methods can be adapted to detect PPIs in almost all cellular compartments, provided the FP used is able to fluoresce also in oxidizing conditions. The SIMPL method presented here is, also in this respect, not the solution to the problem, since, in my opinion it would fail for proteins in oxidizing compartments such as ER and mitochondria because inteins contain cysteines which would form disulfide bonds in oxidizing compartments (see also comment below about the experiment in mitochondria).

Beyond these points, my strongest criticism is that I think inteins are not suited for this type of assay. Inteins are difficult to work with because they are poorly folded (the C-intein being completely disordered and the N-intein being partially disordered; Shah et al., *JACS*, 2013). When fused to other proteins, they often lead to poor expression of the fusion proteins. Moreover, the splicing reaction is entirely dependent on the amount of N- and C-intein fragments. Changes in expression levels of either the bait or prey constructs would lead to very different results, compromising the reliability of the conclusion regarding the interaction. Here the authors had to split the intein at a different position to decrease the affinity between the two fragments, which otherwise would give rise to a huge background. This also leads to lower splicing efficiency. This likely compromises the ability to detect PPIs at physiological protein expression levels.

Taken together these considerations, I do not feel it appropriate for this paper to be published in *Nature Communications*.

Below are further comments to the authors.

1. The authors should directly start by saying that they had to mutate the intein in order to have little to no affinity between the two fragments. It is highly misleading otherwise, because inteins are used in bioengineering to bring together proteins/peptides that have nothing to do with each other! I think that experiments with the wild type intein should be omitted or used as negative control, but cannot by any means represent the first dataset.
2. Decreasing the affinity between intein fragments has been done in the past to control the trans-splicing reaction with light (Tyszkiewicz and Muir, *Nature Methods*, 2008).
3. Inteins are small, this is true, but they are highly problematic when it comes to folding issues.

Even when fusing them to otherwise properly folded proteins they create problems and lead to very low expression levels of the fusion proteins. The authors should mention this problem as several PPIs may not be detected because the prey and bait (one or both) may not be expressed properly when in fusion with the intein fragments. This can be seen directly in the data shown in Fig. 1c, where it is clear that the FRB-IntN fusion is poorly expressed in comparison to the IntC-FKBP1A fusion.

When expressed together, the intein fragments can bind and this stabilizes the proteins. I think this is the reason why the authors get such a large amount of spliced product when using rapamycin. Indeed, since the two intein fragments work in a 1:1 ratio, the spliced product can be only as much as the least abundant between the IntN and IntC. The authors should discuss this in the paper, as it may be very confusing for readers otherwise.

4. In the Western blots, loading controls should be shown. This is standard practice. Moreover, in this specific case, it is very important to convince the readers that the spliced product is not visible in the absent of interaction not because less sample was loaded on the gel, but because the splicing did not occur.

5. I had a hard time deciphering the schematics. I strongly suggest the authors to simply represent their constructs with the various components from N- to C-terminus. There is no problem in writing something like: V5-IntN-Bait.

6. In Fig. 2b, I cannot reconcile the bands shown for the NIN-FRB+IC-FKBP1A sample with what they should be. In my opinion, the IN and NIN constructs should have the same size, and this should approx. be 22 kDa. Thus, in the lane in which only NIN-FRB is shown, the band should run below 25 kDa, just like it does for the IN-FRB sample, shown in the first lane. Same problem for the spliced product: according to my calculations, it should be approx. 15 kDa (indeed, Bait-V5-FLAG and Prey-V5-FLAG should be very similar in size, thus the band corresponding to the product should run at a very similar size as the product for the FRB-IN+FKBP1A-CIC). For this calculation, I considered the following sizes: FRB: 11 kDa, IN (100 aa): 11 kDa, IC (25 aa): 3 kDa, FRB-IN: 22 kDa, FKBP1A: 12 kDa, IC-FKBP1A: 15 kDa, FRB-FKBP1A (considering also the various linkers, which however should not add more than a couple kDa to the entire construct): 26-27 kDa.

7. I like the fact that the authors thought of problems that might arise when tagging a protein of interest to the "wrong" terminus. However, eventually they still did not consider all possible scenarios: what if the bait tolerates only N-terminal tagging and the prey only C-terminal tagging? The problem of the small size of the spliced product (only the V5+FLAG tags according to the current design) could be easily overcome by adding some other protein instead of the tags, be it GFP, MBP or TRX.

I think the authors should add one such scenario to the paper for completeness.

8. In Fig. 3c: the blue curve seems to suggest that there is quite some signal in the absence of rapamycin. Would that mean that one would actually think the bait and prey interact when in fact they don't? Indeed, for me this is a critical point about the paper: split inteins really like to interact and, despite the fact that the new splice site reduced the affinity of the two inteins (and also their splicing ability, judging by the amount of N- and C-fragments still remaining on the blots in all cases...), I still believe the background is non-negligible and when expression levels are high enough, false positives may be a true problem. The authors, therefore, should show what kind of signal they get in the absence of prey all together.

9. In the ELISA set-up: the authors normalize their signal after splicing to the signal of the bait. However, splicing will be affected by the amount of both, prey and bait. If in the various conditions (e.g. different rapamycin concentrations), different amount of prey is used, the final readout could be very misleading.

As far as I understood from the method section, in Fig. 3c cells were independently transfected

with the constructs and treated with the different doses of rapamycin. My concern is that transfections lead to very inhomogenous expression levels. Not knowing the amount of the prey is a problem somehow, because it makes it difficult to judge the final product.

Splicing is highly dependent on the concentrations of the intein fragments...

Indeed, what are the physiological levels of the various proteins tested in this paper? Would the interaction be detected were the levels very low?

On the opposite hand of the spectrum: what if bait and prey are both expressed at too high levels? The authors should show titrations of their constructs and the resulting splicing.

10. In Fig. 3f: did all bait give a signal? Or did the authors encounter expression problems in some cases?

Moreover, is a PPI considered positive if positive in any of the other methods or in all of them?

11. Fig. 4a, b: also here, loading controls must be included to convince the readers that splicing did not occur in the absence of interaction.

12. Fig.4c,d: The authors emphasize that their method is able to detect PPIs in various cellular compartments. However, I do have some difficulties in believing that the splicing effectively occurred in the mitochondria, because of their oxidizing nature. Inteins have been shown not to work under oxidizing conditions due to the formation of disulfide bonds (Aranko et al., PLoS ONE, 2009). Therefore, what I think it's happening is that the reaction is taking place after cell lysis. Indeed, the data shown here are from an ELISA assay, that is, cells were lysed, thus compartmentalization was lost. In Material and Methods, it is mentioned that cell lysates were incubated for 3h at 4C. I wonder whether in this time, the intein fragments had the time to meet and react, thus the spliced product would not necessarily reflect an interaction that occurred inside the compartment. I suggest the authors to do this experiment: they should mis-localize one of the baits or preys, for instance by adding an NES and having the protein in the cytosol. The corresponding partner should instead reside in the inner mitochondrial membrane or the matrix, as normal. Splicing should not occur in this case, unless it happens when the proteins meet after cell lysis.

Minor points

1. In Fig. 1d and e, HEK 293 should be written on top of the image to be consistent with the other panels in the same figure.

2. In Fig. 1f and g, the molecular weights on the side of the image are missing.

Reviewer #2:

Remarks to the Author:

Yao et al. describe a clever method termed SIMPL to detect protein-protein interactions by fusing two proteins to the N- and C-terminal fragments of an intein. Binding of the two proteins reconstitutes the intein, which results in a splicing event that fuses the two proteins into a single polypeptide. This (or another fused) polypeptide can be detected by a mobility shift on a Western blot. Alternatively, an ELISA assay can be used that takes advantage of two epitope tags that become fused together after intein activity, one for protein capture and one for detection by an antibody coupled to horseradish peroxidase.

While the method is likely to prove useful, the manuscript has a couple of major limitations. One,

the data presented largely consist of testing known interactions and comparing their detection to several other methods. Some of the comparison methods are applicable, like SIMPL, only to defined protein pairs that can be used to construct tagged versions, while others, like yeast two-hybrid or protein complementation assays, can be applied to detect previously unknown interactions using library screens. Thus, given that the manuscript's most common readout for defined protein pairs is a Western blot – a non-trivial amount of work – and in the absence of novel results with the SIMPL assay, it is difficult to judge whether the method will achieve broad acceptance and usage.

The second concern is the lack of, or unclear, quantification of the results in the manuscript. Signals on Western blots are described with terms like "relatively high level," "hardly detected," "no obvious signal," "effectively detected," "successfully detected" etc. For an experimenter attempting this method and observing weak bands in the spliced position that is indicative of protein interaction, this kind of terminology is unlikely to be of much use in determining if an interaction has occurred. Can some kind of standard be applied and defined for users?

In the case of the ELISA readout, for which quantification is provided, the manuscript can be cryptic. Supplemental Table 2 provides specific quantitative data, but these are aggregated in Figure 3f as "positive" or "negative." More useful in Figure 3f might be a heatmap that indicates the relative degree of signal. It would also be helpful to have the ELISA results roughly calibrated to signals on the Western blots. Further, in Figure 3c, one of the arrangements of the intein tags (red line) maxes out in luminescence upon protein interaction (highest level of rapamycin) near the basal level without an interaction (no rapamycin) of one of the other arrangements (purple line). How does one decide which arrangement of intein domains to use without testing multiple combinations?

The data for the high AUC values for the reference datasets of protein interactions for the SIMPL assay are not sufficiently described. For example, "positive" SIMPL results range from RLU means of 3.97 down to 0.58, with about half of these positives having a value <1.0. An additional 14 pairs, 6 of which are labeled as false positives, are at least 80% of the low end 0.58 value. Here again, a user of the assay who will not want to reconstitute and retest the whole reference set of data would struggle to interpret intermediate RLU values. I could find no clear statement in the text of what constitutes "detection" of an interaction by SIMPL.

In Supplemental Figure 3, many RLU values are shown as positive SIMPL signals, all the way down to <0.125. Is this the same scale as Supplemental Table 2? Are these values provided after subtraction of background?

Minor concerns

One concern is that the manuscript is more promotional than might be appropriate. I found statements like "SIMPL displays an excellent ability to follow the kinetics..."; "these results clearly demonstrate the potential for using SIMPL as a highly sensitive...tool"; "SIMPL is an exceptionally sensitive method"; "an unprecedentedly high sensitivity" etc. to be opinions of the authors that might be toned down in favor of more objective statements.

What is a "regular" protein detection method (Abstract and line 124).

Fig. 2a could be simplified, especially the inverted letters labeling the domains, to better convey the arrangements.

Response to Reviewer #1

Comment 1: *In this paper, Yao and colleagues report the development of a method to detect protein-protein interactions (PPIs) based on the trans-splicing reaction mediated by the split intein gp41-1. The authors justify why they decided to develop such method saying that the current methods have several limitations, for instance the inability to detect weak and transient interactions or to function in all cellular compartments. While I do agree that good methods to robustly detect PPIs are very important, I fail to see how the method presented here represents the solution to the problem.*

Response to Comment 1: While there are many different PPI detection methods available, we understand that all have their individual strengths (and weaknesses). We did not mean to imply that SIMPL methodology has no such limitations or provides a complete solution to many of the challenges facing proteomics screening, and we regret not expressing this more clearly. Rather, we were trying to emphasize the robustness of our assay, which we demonstrated, in our ROC analyses (Fig. 3d-f), showed better overall performance than many other modern and classical approaches using a well-defined control set of reference proteins. This is achieved through its ability to detect weak and transient PPIs and PPIs in many different cellular locations, although we recognize that SIMPL might not be the best in each aspect. We also acknowledge that our comparison, which was based on available data extracted from the literature, cannot cover all available methodologies, due to a lack of similar benchmarking evaluation for these approaches. We do feel that our results clearly demonstrate the strengths of SIMPL, however, and show how it could be used in conjunction with other approaches to provide a more comprehensive coverage of the protein interaction space. We apologize for any confusion and have changed the tone and content of our description in the revised Abstract, Introduction and Discussion.

Comment 2: *Weak and transient interactions can be detected to date either using complementation of fluorescence (BiFC) or transcriptional readouts. In both cases, transient interactions are converted in a long-lasting response (either fluorescence which, unless specifically engineered to be labile, is quite stable, or a gene product). Few years ago, a method called SPARK was developed that combines a transcriptional readout with light activation, so that even PPI dynamics can now be tracked. The authors do not cite the SPARK method (Kim et al., *elife*, 2017), however in my eyes this method is more powerful than the one described here. It is also compatible with high-throughput and can be used for drug screening.*

Response to Comment 2: We agree with the reviewer that there are several methods such as BiFC and SPARK capable of detecting weak and transient PPIs and did not mean to imply that such approaches are unavailable. However, the time scale for BiFC is generally in the order of hours, which makes detecting transient PPIs not a strength of BiFC. Transcription readout-based methods are also usually not very suitable for detecting transient interactions since they require long term binding to drive downstream transcription. We recognize that SPARK is an excellent technique and cleverly overcomes several general problems by employing a light-controlled device which allows time-gated information flow to produce signal and, as suggested by the reviewer, we have cited SPARK in the Discussion of our revised manuscript. Notably, upon examination of the data presented in the paper (Kim et al, *eLife*, 2017) we found that SPARK and SIMPL display similar performance, with both systems detecting rapamycin-induced FRB/FKBP interaction within 2 min. However, SPARK seems limited mainly to

membrane PPIs, while the SIMPL assay is useable across a much more diverse set of cellular localizations. We also acknowledge that other methods such as FRET and BRET, and in particular the newly developed nanoBRET, are excellent tools for detecting transient interactions. Large scale comparison of all of these approaches in the future could therefore certainly be useful, and we fully realize that other methods may offer advantages over SIMPL in different cases. However, direct comparison to many other proteomics assays is a significant undertaking well beyond the scope of this work. Rather our goal in this manuscript is to illustrate the strength and versatility of SIMPL, not to try and establish it as a superior approach in all circumstances. We believe that the modified tone of our revised manuscript better reflects this and the revised Discussion more clearly addresses the capabilities of other available technologies.

Comment 3: *BiFC-based methods can be adapted to detect PPIs in almost all cellular compartments, provided the FP used is able to fluoresce also in oxidizing conditions. The SIMPL method presented here is, also in this respect, not the solution to the problem, since, in my opinion it would fail for proteins in oxidizing compartments such as ER and mitochondria because inteins contain cysteines which would form disulfide bonds in oxidizing compartments (see also comment below about the experiment in mitochondria).*

Response to Comment 3: We agree with the reviewer that BiFC is also a good method for monitoring PPIs in different cellular compartments. However, the readouts between SIMPL and BiFC are distinct, and we believe researchers can only benefit from having more diverse methods with similar capabilities available.

We also thank the reviewer for expressing their concerns about oxidation and agree that the crucial first amino acid of the GP41-1 split intein is oxidation-labile cysteine. Notably, oxidation has been one of our concerns while building the SIMPL system from the beginning. We first examined this by testing the GP41-1 reaction in the extracellular space, which is the most oxidative environment. To accomplish this, an interacting pair of transmembrane proteins was IN and IC tagged such that the inteins were located in the extracellular space. These constructs were then transfected into different cells and, after time to allow for protein expression, the cells were mixed and subject to immunoprecipitation. We observed that the WT GP41-1 split intein tags underwent splicing in this configuration (i.e. in the oxidative environment of the extracellular space) although the efficiency was markedly reduced (Fig. A1 as below). Splicing was also able to occur along the sorting pathway composed of the ER and endosome, which have relatively higher redox potential than other organelles (but lower than in the extracellular space), as shown by coexpression of the tagged proteins in the same cells (Fig. A1). Therefore, we conclude that the GP41-1 split intein is relatively resistant to oxidation compared to other inteins studied previously. We reason that the resistance might be partly derived from the fact that the key first amino acid of the C-terminal extein is serine instead of cysteine, although a quantitative study of the redox impact of specific residues in the GP41-1 split intein is beyond the scope of this study. Ultimately, the most robust SIMPL solution for studying PPIs in oxidative environments would be to rebuild the system with a non-Cys (i.e. Ser/Thr) based split intein, a process which is currently underway in our lab. We have, however, shown that our current configuration still does function under relatively oxidizing conditions, and is therefore a useful tool for studying diverse types of interactions.

Additionally, we would like to note that, in contrast to other organelles, the mitochondrion actually has the lowest redox potential in many cell lines although it is most susceptible to oxidative damage (BBA 1780: 1273-1290). Thus, the difficulties of studying mitochondrial PPIs usually do not result from the redox environment but mainly from the distinct protein sorting mechanisms and the special local structure of the organelle, which can be largely overcome by the SIMPL strategy shown in Fig. 4d.

Appended figure A1. Assessment of extracellular performance of GP41-1 split intein. **(a)** Plasmids for extracellular IN/IC tags containing signal sequence-IC-FLAG-PD1 or signal sequence-V5-IN-PDL1 (WT IN and IC in this case) were separately transfected into HEK 293 cells in different wells. After protein expression, the cells were resuspended in PBS and the two pools of cells were mixed together at 37°C for different periods of time. The cells were collected, lysed and subject to immunoprecipitation with α -FLAG Ab. **(b)** The precipitants were subject to immunoblot analysis. Spliced protein was precipitated with α -FLAG Ab and is recognizable by α -V5 Ab as highlighted with red arrows. Due to the spatial restriction, splicing should occur outside cells (in sample of separate expression) or along the secretion pathway (in sample of co-expression). Heavy chain and light chain from immunoglobulin are highlighted with white asterisk. c.e.: coexpression.

Comment 4: Beyond these points, my strongest criticism is that I think inteins are not suited for this type of assay. Inteins are difficult to work with because they are poorly folded (the C-intein being completely disordered and the N-intein being partially disordered; Shah et al., JACS, 2013). When fused to other proteins, they often lead to poor expression of the fusion proteins. Moreover, the splicing reaction is entirely dependent on the amount of N- and C-intein

fragments. Changes in expression levels of either the bait or prey constructs would lead to very different results, compromising the reliability of the conclusion regarding the interaction. Here the authors had to split the intein at a different position to decrease the affinity between the two fragments, which otherwise would give rise to a huge background. This also leads to lower splicing efficiency. This likely compromises the ability to detect PPIs at physiological protein expression levels.

Response to Comment 4: We thank the reviewer for providing his/her insights into the potential difficulties of using split inteins. However, our data strongly demonstrate that split intein can be reliably harnessed as a molecular sensor of PPIs in the SIMPL system. In terms of folding, although a disordered conformation can potentially lead to fast protein degradation, fusing a disordered peptide to a protein does not necessarily lead to poor expression. Indeed, the split sensors in most PCA methods, including BiFC, are similarly unfolded before complementation. In the specific case of the re-split GP41-1 intein, we have typically observed decent expression of fused proteins, although the expression level does vary as shown in revised Supplementary Fig. 3a. This variation can be caused by many factors in addition to tagging, however. Also, while we do not completely exclude the possibility of the intein tag inducing degradation in some cases (which could lead to false negatives), this does not appear to be a frequent occurrence, as evidenced by our ROC analysis presented in Fig. 3d-e, which demonstrates excellent performance of SIMPL across a variety of PPI types. Additionally, we view the unfolding of the intein fragments as a benefit to facilitate the SIMPL system: before reconstitution, the unfolded or partially folded states prevent the split intein from deleterious side reactions; upon binding, the two fragments undergo a disorder-to-order transition to fold into a HINT structure which allows splicing. Note that a more thorough discussion of the folding issue has been incorporated into the Discussion session of the revised manuscript.

In terms of intein affinity and splicing efficiency concerns, it is important to point out that our reengineered split intein had most, if not all, of its intrinsic affinity abolished, but did not appear to have compromised activity as shown in revised Fig. 1c which clearly indicates its activity (C25) is comparable to that of WT (C37). (Note that for clarity we have repeated our experiment with the different split intein forms and present it in the revised manuscript as a single panel collectively showing all of the re-engineered split inteins together). These data suggest that our C25 GP41-1 split intein therefore avoids the problem of the 'splicing reaction being entirely dependent on the amount of N- and C- intein fragments' as stated by the reviewer (though we agree that with WT intein this would potentially be an issue). Thus splicing is not driven by the concentration of C25 GP41-1, but instead by extrinsic affinity between the attached proteins. Spontaneous splicing only occurs when the expression is extremely high as shown in revised Supplementary Fig. 5 (RRS panel). Note that we have numerous examples showing that specific interactions, not the expression itself, induce splicing, such as the time course of rapamycin-induced FRB/FKBP interaction (revised Fig. 1e and Fig. S2c-d), the effects of expression level on rapamycin-inducing FRB/FKBP splicing (revised Fig. 1f) and EGF-induced EGFR/SHC1 splicing (revised Fig. 4c). The capability of SIMPL to follow fast interactions as short as 2 minutes further proves the efficiency of C25 GP41-1. Therefore we don't believe that our efforts to reduce the intrinsic affinity of the intein fragments has had an adverse effect on intein activity, and we also don't believe that complete inhibition of this intrinsic activity is even necessary for assay function. In addition, basal splicing can be distinguished from interaction signal by quantitative assay such as ELISA as demonstrated in our evaluations with reference sets (revised Fig. 3d-f and Fig. 5), or by controlling expression level as demonstrated in revised

Fig. 1f and Fig. 4c. It should be noted that independent sensor fragment association is a severe and common problem for some PCA assays such as BiFC but does not impede them as well accepted methods. What is important is a proper balance between reduced association and activity, which has been achieved with the C25 GP41-1 intein.

With respect to the effects of disproportionate bait/prey expression we agree that the splicing reaction occurs at a 1:1 ratio, however this is not different from other PCA assays including BiFC. A potential problem specific to split inteins is side reactions such as terminal cleavage. Our Western blot data presented in Fig. 1, 2, 4, 6, S2 and S4, however, clearly show that the spliced proteins are the major bands, excluding the contamination of side reactions even when bait or prey proteins are expressed alone. While trace amounts of bands with unexpected molecular weight did appear in some samples, these bands should not skew SIMPL signals with their low stoichiometry.

Comment 5: *The authors should directly start by saying that they had to mutate the intein in order to have little to no affinity between the two fragments. It is highly misleading otherwise, because inteins are used in bioengineering to bring together proteins/peptides that have nothing to do with each other! I think that experiments with the wild type intein should be omitted or used as negative control, but cannot by any means represent the first dataset.*

Response to Comment 5: We thank the reviewer for this valuable suggestion and have modified the manuscript accordingly. Accordingly, the original Supplementary Fig. 2a has been removed.

Comment 6: *Decreasing the affinity between intein fragments has been done in the past to control the trans-splicing reaction with light (Tyszkiewicz and Muir, Nature Methods, 2008).*

Response to Comment 6: We thank the reviewer for introducing us to this example which demonstrates previous efforts to reduce basal interaction in splicing systems. However, the work was done in a different background and used a different technical strategy, and therefore does not compromise the novelty of the re-split GP41-1 intein. More specifically, in this earlier paper, a split intein (VMA) serves as an actuator under the control of a sensor, a light-induced dimerization pair (PhyB/PIF3), in a conditional protein splicing (CPS) system. The authors of this work re-engineered the sensor instead of the split intein to reduce the basal affinity of the whole system. While intriguing, this approach is unfortunately not generally applicable in PPI studies or other sensor mediated CPS. It should be noted that the split intein used in this system, VMA, has a very slow reaction rate. This may explain the low background signal in the CPS system, however the system showed correspondingly slow kinetics even though driven by light illumination (Fig. 3 in the paper). In contrast, we re-engineered the sensor split intein directly in the SIMPL system, introducing features (described above) that may even have broad applications beyond PPI detection.

Comment 7: *Inteins are small, this is true, but they are highly problematic when it comes to folding issues. Even when fusing them to otherwise properly folded proteins they create problems and lead to very low expression levels of the fusion proteins. The authors should mention this problem as several PPIs may not be detected because the prey and bait (one or both) may not be expressed properly when in fusion with the intein fragments. This can be seen directly in the data shown in Fig. 1c, where it is clear that the FRB-IntN fusion is poorly expressed in comparison to the IntC-FKBP1A fusion. When expressed together, the intein*

fragments can bind and this stabilizes the proteins. I think this is the reason why the authors get such a large amount of spliced product when using rapamycin. Indeed, since the two intein fragments work in a 1:1 ratio, the spliced product can be only as much as the least abundant between the *IntN* and *IntC*. The authors should discuss this in the paper, as it may be very confusing for readers otherwise.

Response to Comment 7: As mentioned above, we have provided a more through discussion of potential folding issues in Discussion of the revised manuscript. Also, as answered in Response to Comment 4, our data do not support the assumption that split intein-induced low protein expression due to folding state is a major issue in SIMPL system. We agree with the reviewer that the expression of FRB (IN) fused protein in Fig. 1c is relatively lower than FKBP1A (IC), and that this could be a result of its fusion with an unfolded tag. However, this is not the only possible cause; as FRB is a small fragment derived from a large protein mTOR, its own structure may not be stable when it is expressed without other associated proteins. This assumption is supported by the fact that binding of rapamycin increased its expression when it is not accompanied by FKBP1A expression as seen in Fig. A2 (below). Please note that an unrelated protein, IN-fused Ras binding domain fragment derived from RAF1 (RBD (IN)) was not affected by rapamycin. Therefore, the stability of FRB itself might be the main factor affecting the expression level of FRB (IN). With respect to the stoichiometry of the SIMPL reaction, it does not represent the real dynamics of PPIs but should reflect the association rate, and therefore provides important information on the target PPIs. This issue has been addressed in the Discussion. It should be noted that all PCA systems undergo 1:1 ratio reactions without exception, and their results should be also interpreted carefully.

Appended Figure A2. FRB (IN) and RBD (IN) were expressed alone or together in HEK 293 cells. After treatment with different concentrations of rapamycin as indicated for 2 hours, the cells were lysed and subject to Western blot analysis with α -V5 antibody.

Comment 8: *In the Western blots, loading controls should be shown. This is standard practice. Moreover, in this specific case, it is very important to convince the readers that the spliced product is not visible in the absent of interaction not because less sample was loaded on the gel, but because the splicing did not occur.*

Response to Comment 8: We appreciate the reviewer's criticism and have added loading controls to the figures of the revised manuscript.

Comment 9: *I had a hard time deciphering the schematics. I strongly suggest the authors to simply represent their constructs with the various components from N- to C-terminus. There is no problem in writing something like: V5-IntN-Bait.*

Response to Comment 9: We thank the reviewer for this helpful suggestion and modifications have been made accordingly in the Results and Fig. 2-3 of the revised manuscript.

Comment 10: *In Fig. 2b, I cannot reconcile the bands shown for the NIN-FRB+IC-FKBP1A sample with what they should be. In my opinion, the IN and NIN constructs should have the same size, and this should approx. be 22 kDa. Thus, in the lane in which only NIN-FRB is shown, the band should run below 25 kDa, just like it does for the IN-FRB sample, shown in the first lane. Same problem for the spliced product: according to my calculations, it should be approx. 15 kDa (indeed, Bait-V5-FLAG and Prey-V5-FLAG should be very similar in size, thus the band corresponding to the product should run at a very similar size as the product for the FRB-IN+FKBP1A-CIC). For this calculation, I considered the following sizes: FRB: 11 kDa, IN (100 aa): 11 kDa, IC (25 aa): 3 kDa, FRB-IN: 22 kDa, FKBP1A: 12 kDa, IC-FKBP1A: 15 kDa, FRB-FKBP1A (considering also the various linkers, which however should not add more than a couple kDa to the entire construct): 26-27 kDa.*

Response to Comment 10: We apologize for the unclear description. In the NIN and IN constructs, two HA tags are included. We also include a Myc tag in the IC and CIC constructs (although this tag was not used in our experiments due to its poor performance in this context). There are also some additional amino acids present in the constructs that are derived from the Gateway cloning site. The arrangements and exact molecular weights of the constructs and their products are listed below. Sequence information has been added as Supplementary Table 1 in the revised manuscript.

FRB (IN): FRB-HAx2-V5-EN-IN, 29.5 kDa

FRB (NIN): HAx2-V5-EN-IN-FRB, 30.7 kDa

FKBP1A (IC): IC-EC-FLAGx3-Myc-FKBP1A, 22.4 kDa

FKBP1A (CIC): FKBP1A-IC-EC-FLAGx3-Myc, 21.8 kDa

Product of FRB (IN) + FKBP1A (IC): FRB-HAx2-V5-EN-EC-FLAGx3-Myc-FKBP1A, 37.6 kDa

Product of FRB (IN) + FKBP1A (CIC): FRB-HAx2-V5-EN-EC-FLAGx3-Myc, 22.9kDa

Product of FRB (NIN) + FKBP1A (IC): HAx2-V5-EN-EC-FLAGx3-Myc-FKBP1A, 24.1 kDa

Please note that the bands in the Fig. 2b of the original manuscript may have been skewed to some extent due to a technical issue, although their relative positions were as expected. We repeated the experiment together with FRB (NIN)/FKBP (CIC-GFP) and have presented a revised Fig. 2b, where the bands are more in agreement with the expected molecular weights. Note that the image provided is representative of three repeated experiments, all showing consistent results/sizes.

Comment 11: *I like the fact that the authors thought of problems that might arise when tagging a protein of interest to the “wrong” terminus. However, eventually they still did not consider all possible scenarios: what if the bait tolerates only N-terminal tagging and the prey only C-terminal tagging? The problem of the small size of the spliced product (only the V5+FLAG tags according to the current design) could be easily overcome by adding some other protein instead of the tags, be it GFP, MBP or TRX. I think the authors should add one such scenario to the paper for completeness.*

Response to Comment 11: We thank the reviewer for the suggestion. We have now created a CIC-GFP construct and studied its performance in combination with NIN bait as suggested. The data has been included in revised Fig. 2-3.

Comment 12: *In Fig. 3c: the blue curve seems to suggest that there is quite some signal in the absence of rapamycin. Would that mean that one would actually think the bait and prey interact when in fact they don't? Indeed, for me this is a critical point about the paper: split inteins really like to interact and, despite the fact that the new splice site reduced the affinity of the two inteins (and also their splicing ability, judging by the amount of N- and C-fragments still remaining on the blots in all cases...), I still believe the background is non-negligible and when expression levels are high enough, false positives may be a true problem. The authors, therefore, should show what kind of signal they get in the absence of prey all together.*

Response to Comment 12: We agree that the NIN construct sometimes displays relatively high basal signal. Consistent with this, FRB (NIN)/FKBP1B also produced high basal signal as presented in our revised Fig. 3c. It is unlikely that this signal is solely derived from the self-association of C25 GP41-1 intein, however, because it was not observed with the IN format. We reason that other factors such as surrounding sequence may also contribute. Despite this fact, the FRB (NIN)/FKBP1A (IC) and FRB (NIN)/FKBP1A (CIC-GFP) combinations still demonstrate nice response to rapamycin which proves the usefulness of NIN construct if proper controls are used. We also assessed bait alone as suggested by the reviewer and the data have been included in revised Fig. 3c. Therefore, without FKBP1A prey, neither FRB (IN) nor FRB (NIN) demonstrated any response to rapamycin.

Comment 13: *In the ELISA set-up: the authors normalize their signal after splicing to the signal of the bait. However, splicing will be affected by the amount of both, prey and bait. If in the various conditions (e.g. different rapamycin concentrations), different amount of prey is used, the final readout could be very misleading. As far as I understood from the method section, in Fig. 3c cells were independently transfected with the constructs and treated with the different doses of rapamycin. My concern is that transfections lead to very inhomogenous expression levels. Not knowing the amount of the prey is a problem somehow, because it makes it difficult to judge the final product. Splicing is highly dependent on the concentrations of the intein fragments...*

Indeed, what are the physiological levels of the various proteins tested in this paper? Would the interaction be detected were the levels very low? On the opposite hand of the spectrum: what if bait and prey are both expressed at too high levels? The authors should show titrations of their constructs and the resulting splicing.

Response to Comment 13: We agree with the reviewer that normalization to both bait and prey should theoretically provide more reliable signal. However, in practice it usually introduces more error from the different measurements, skewing the results and making the data difficult to interpret. Note that this is not confined simply to our assay; significantly, there are no PPI methods, to the best of our knowledge, that use a double normalization strategy. We actually made an effort to do this by including another tag, Myc, in our prey constructs. However, the performance of Myc tag in this context was far from satisfactory. We are attempting to improve on this in future studies using different tags. However, we envision that the normalization will not be realized simply by division by both bait and prey levels, but by new algorithms.

We also agree with the reviewer that variance in the expression of bait and prey may be an issue and potentially skew the results to some extent. However, transfection approaches are still valuable and commonly used by most methods including PCA and SPARK. The evaluation of

SIMPL with reference sets (Fig. 3d-f) was also performed using a transfection-based strategy and the overall performance proved the validity of this approach. To better address the reviewer's concerns, however, as well as obtain more insight into the SIMPL system, we created some stable cell lines with both bait and prey genes incorporated into the host genome and their expression under the control of the Tet operator (revised Fig. 1f and 4c). By this means more consistent expression of bait and prey is possible. As presented in Fig. 1f, a wide range of FRB/FKBP1A expression was achieved by titration with different dosages of tetracycline, including an extremely low level just detectable by immunoblot. Conversely, to achieve maximal expression, we transfected cells with the expression plasmids. Without exception, splicing was observed in all samples with rapamycin treatment. In contrast, no significant signal was observed in non-treated cells. These results demonstrated the wide dynamic range of the SIMPL assay. Similarly, EGF-stimulated EGFR/SHC1 interaction can also be detected by SIMPL using low bait/prey concentration as presented in Fig. 4c. Please note that in this figure there is low level of leaky expression without tetracycline treatment, however we presented a blot with long exposure to allow visualization of splicing signal. Additionally, it should be noted that the samples were treated with EGF for only two minutes, again indicating the sensitivity and fast kinetics of SIMPL assay. Unfortunately, we were unable to lower EGFR expression to its endogenous level due both to the leaky expression of the protein and the extremely low levels of EGFR natively expressed in HEK 293 cells. In the future we will examine other proteins whose expression can be manipulated to near physiological levels. However, we don't believe that detecting PPIs at a physiological level is a strength of SIMPL (or other PCA methods or SPARK). Rather, we aim to develop SIMPL as a suitable tool for PPI discovery. More detailed characterization of PPIs at physiological levels should be carried out by alternate methods such as PLA.

Comment 14: *In Fig. 3f: did all bait give a signal? Or did the authors encounter expression problems in some cases? Moreover, is a PPI considered positive if positive in any of the other methods or in all of them?*

Response to Comment 14: To address the concern, we plot bait expression in the revised Supplementary Fig. 3a. As shown in the figure, most of the proteins were expressed above basal level, although expression level does vary. Only a small portion of proteins exhibited signals close to the basal line, and these included proteins involved in cell death which may induce low signal via cell death. The positive reference set was adopted from a previously published work (Nature Methods 6:91-97) and was designed from well documented PPIs. We used ROC analysis to calculate the cut-off value and used the cut-off value to determine positives: readings above the cut-off value are considered positive. This is a well accepted statistical method also used by other studies, and allowed us to better compare our approach with other methods. Comparison of the results of different methods in detecting specific PPIs is presented in Fig. 3f. Notably, methods vary in their ability to detect different PPIs, with only a few PPIs being detected by all (or none) of the approaches. It should be noted that SIMPL exhibits the broadest coverage in this test.

Comment 15: *Fig. 4a, b: also here, loading controls must be included to convince the readers that splicing did not occur in the absence of interaction.*

Response to Comment 15: We thank the reviewer and loading controls have been added in revised Fig. 4a-b.

Comment 16: *Fig.4c,d: The authors emphasize that their method is able to detect PPIs in various cellular compartments. However, I do have some difficulties in believing that the splicing effectively occurred in the mitochondria, because of their oxidizing nature. Inteins have been shown not to work under oxidizing conditions due to the formation of disulfide bonds (Aranko et al., PLoS ONE, 2009). Therefore, what I think it's happening is that the reaction is taking place after cell lysis. Indeed, the data shown here are from an ELISA assay, that is, cells were lysed, thus compartmentalization was lost. In Material and Methods, it is mentioned that cell lysates were incubated for 3h at 4C. I wonder whether in this time, the intein fragments had the time to meet and react, thus the spliced product would not necessarily reflect an interaction that occurred inside the compartment. I suggest the authors to do this experiment: they should mis-localize one of the baits or preys, for instance by adding an NES and having the protein in the cytosol. The corresponding partner should instead reside in the inner mitochondrial membrane or the matrix, as normal. Splicing should not occur in this case, unless it happens when the proteins meet after cell lysis.*

Response to Comment 16: As we mentioned above, the mitochondrion actually has the lowest redox potential in many cell lines. However, the reviewer made excellent points and we acknowledge the importance of excluding the possibility of splicing during sample processing. We employed two approaches for this purpose using PDHA1/PDHB interaction as an example (please see revised Fig. 4f). Firstly, similar to what suggested by the reviewer, we replaced the targeting sequence of PDHA1 with an NLS (thereby preventing its sorting to the mitochondrion) and assessed its interaction with PDHB using SIMPL. Secondly, we expressed PDHA1 and PDHB separately and performed ELISA by mixing lysates from the two samples. Both cases did not show any signal above background. We are therefore confident in concluding that the splicing we reported occurred properly in the mitochondrion.

Comment 17: *In Fig. 1d and e, HEK 293 should be written on top of the image to be consistent with the other panels in the same figure.*

Response to Comment 17: We moved these two figures to the Supplementary Data (Supplementary Fig. 2c-d) in the revised manuscript and have labelled them accordingly.

Comment 18: *In Fig. 1f and g, the molecular weights on the side of the image are missing.*

Response to Comment 18: Marks indicating molecular weight have been added.

Response to Reviewer #2

Comment 1: *While the method is likely to prove useful, the manuscript has a couple of major limitations. One, the data presented largely consist of testing known interactions and comparing their detection to several other methods. Some of the comparison methods are applicable, like SIMPL, only to defined protein pairs that can be used to construct tagged versions, while others, like yeast two-hybrid or protein complementation assays, can be applied to detect previously unknown interactions using library screens. Thus, given that the manuscript's most common readout for defined protein pairs is a Western blot – a non-trivial amount of work – and in the*

absence of novel results with the SIMPL assay, it is difficult to judge whether the method will achieve broad acceptance and usage.

Response to Comment 1: In the current proof-of-principle study, we developed SIMPL method from scratch and aimed to demonstrate the performance of SIMPL. We believe, although we have not shown this directly in the manuscript, that SIMPL should also be applicable for PPI discovery. There are two ways to achieve this. One is an arrayed screening format, whereby baits of interested are interrogated with prey libraries in a multiwell format (i.e. with each well containing a known bait/prey pair). Throughput of these screens would be determined by the size of the prey library that can be generated and could be facilitated through the use of automated research equipment available in many labs. This format is compatible with the SIMPL – ELISA system described in our manuscript. Another approach would involve screening against a prey library pool, like yeast two hybrid pooled screening. Positive hits would be identified using a selection system followed by sequencing. Theoretically, SIMPL could also be applied in this approach by coupling it to mass spectrometry analysis. Converting SIMPL into a full screening system, however, is beyond the scope of what we are trying to demonstrate with this manuscript, though it would be an excellent subject for future work. In fact, we have recently been awarded a three-year Genome Canada grant that will focus solely on development of SIMPL as a high-throughput screening assay for PPIs and biologics (small molecules and antibodies).

Comment 2: *The second concern is the lack of, or unclear, quantification of the results in the manuscript. Signals on Western blots are described with terms like “relatively high level,” “hardly detected,” “no obvious signal,” “effectively detected,” “successfully detected” etc. For an experimenter attempting this method and observing weak bands in the spliced position that is indicative of protein interaction, this kind of terminology is unlikely to be of much use in determining if an interaction has occurred. Can some kind of standard be applied and defined for users?*

Response to Comment 2: We thank the reviewer for pointing out this issue of poor quantifiability of Western analysis. We believe it is very difficult to set universal standards for Western analysis-based assays. The common practice is to use proper negative and positive controls on a case by case basis. Depending on the nature of the experiments being performed (and their throughput) users may require a more quantifiable approach, however, in which case the SIMPL-ELISA format might be the preferred choice.

Comment 3: *In the case of the ELISA readout, for which quantification is provided, the manuscript can be cryptic. Supplemental Table 2 provides specific quantitative data, but these are aggregated in Figure 3f as “positive” or “negative.” More useful in Figure 3f might be a heatmap that indicates the relative degree of signal. It would also be helpful to have the ELISA results roughly calibrated to signals on the Western blots. Further, in Figure 3c, one of the arrangements of the intein tags (red line) maxes out in luminescence upon protein interaction (highest level of rapamycin) near the basal level without an interaction (no rapamycin) of one of the other arrangements (purple line). How does one decide which arrangement of intein domains to use without testing multiple combinations?*

Response to Comment 3: We are sorry that we did not describe the method more clearly in the previous manuscript. The benchmarking approach is based on receiver operating characterization (ROC) analysis and is accepted by many research labs to evaluate a specific method. In this strategy, a set of well documented PPIs are used as positive reference set (PRS) and a set of protein pairs with low probability of interaction are used as random reference set (RRS). The signals are measured for both PRS and RRS under the same conditions. The positive rates (defined as true positive rate for PRS and false positive rate for RRS) are calculated as functions of varied cut-off values. True positive rates and false positive rates are plotted against one another in an ROC curve as shown in Fig. 3d. The shape of the curve reflects the performance of the assay: better performance if the curve approaches the left-top corner, or poor performance if the curve lies close to the diagonal. From the ROC curve, a cut-off value can be determined that allows optimal differentiation of true and false positives. Using this, pairs above the cut-off value are then considered as positive. Both data presented in Fig. 3e and 3f were calculated according to the analysis in Fig. 3d. The biggest advantage of this analysis is the ability to compare different methods, a major goal of this study. Notably, Fig. 3f reveals a broader coverage achieved by SIMPL versus other methods.

We agree with the reviewer, however, that our representation of this method omitted detail and have added the values of the reads in the revised Supplementary Fig. S3b-c. Also, with the extremely low throughput nature of Western analysis, it is difficult to measure the expression of all baits and preys, which in our case would correspond to 704 samples. However, the example of the rapamycin-induced FRB/FKBP interaction, studied using both Western analysis and ELISA (Fig. 2b and Fig.3c), gives an idea of how the approaches compare.

The differing behavior of the four split intein combinations (including a new combination in the revised manuscript) illustrated in Fig. 3c is primarily derived from the NIN format which has relatively high association with the IC prey. As discussed in our answer to the first reviewer (Response to Comment 12), this is possibly derived from the surrounding amino acid sequences. Ultimately, however, this format still demonstrates good response to rapamycin treatment, so it is usable, though one should be careful and be sure to include appropriate controls.

The choice of tagging arrangement for a target protein should be made according to the protein's own biochemical features. For example, functional or unexposed termini should be avoided for tagging. In the case of unclear biochemical features, all possible combinations should be tested. For PPI screening, it is always better to prepare libraries with different terminal tagging. This discussion has been added to the revised manuscript.

Comment 4: *The data for the high AUC values for the reference datasets of protein interactions for the SIMPL assay are not sufficiently described. For example, "positive" SIMPL results range from RLU means of 3.97 down to 0.58, with about half of these positives having a value <1.0. An additional 14 pairs, 6 of which are labeled as false positives, are at least 80% of the low end 0.58 value. Here again, a user of the assay who will not want to reconstitute and retest the whole reference set of data would struggle to interpret intermediate RLU values. I could find no clear statement in the text of what constitutes "detection" of an interaction by SIMPL.*

Response to Comment 4: As we addressed in Response to Comment 3, we used a well accepted benchmarking method (Nature Methods 6:91-97) to determine positives and negatives. Note that this method is used for evaluation of an assay's performance but not for screening. Positive and negative controls are still needed for PPI screening since variance exists between plates or batches. Additionally, like all screening methods, use of sophisticated statistical algorithms can greatly aid in hit identification and analysis.

Comment 5: *In Supplemental Figure 3, many RLU values are shown as positive SIMPL signals, all the way down to <0.125. Is this the same scale as Supplemental Table 2? Are these values provided after subtraction of background?*

Response to Comment 5: We apologize for the unclear description of this figure (now Supplementary Fig. 3d), which appears to have led to some confusion. They are the same data as in Supplementary Table 2 without subtraction. We were simply plotting the SIMPL readings obtained from the IC format against the readings obtained from the CIC format, as a signal/readout comparison, with no highlighting of whether or not a read is classified as positive or negative. Please note that both axes are in logarithmic scale. This has been explained in the legend for Supplementary Fig. 3d.

Comment 6: *One concern is that the manuscript is more promotional than might be appropriate. I found statements like "SIMPL displays an excellent ability to follow the kinetics..."; "these results clearly demonstrate the potential for using SIMPL as a highly sensitive...tool"; "SIMPL is an exceptionally sensitive method"; "an unprecedentedly high sensitivity" etc. to be opinions of the authors that might be toned down in favor of more objective statements.*

Response to Comment 6: We thank the reviewer and have altered our tone and descriptions to be less promotional in the revised manuscript.

Comment 7: *What is a "regular" protein detection method (Abstract and line 124).*

Response to Comment 7: The "regular" protein detection methods indicate Western blot and ELISA in the Abstract. We have specified this in the revised manuscript. The word "regular" in line 124 has been removed to specify NanoFRET.

Comment 8: *Fig. 2a could be simplified, especially the inverted letters labeling the domains, to better convey the arrangements.*

Response to Comment 8: We thank the reviewer for the suggestion and the figure has been modified in the revised manuscript.

Reviewers' Comments:

Reviewer #1:

Remarks to the Author:

I thank the authors for the thorough explanations that have dissipated my concerns. I think the manuscript is much improved and deserves publication. A side note: recently, a different method, also based on split inteins, has been published that has practically the same name, albeit being written differently: SiMPI (Palanisamy et al., Nature communications, 2019, 10:4967). It could be confusing for the scientific community to have two methods that are used for very different purposes, with practically the same name. I am sure the authors are attached to "SIMPL" as name for their method. However, if they could imagine changing it, it may help.

Reviewer #2:

Remarks to the Author:

I was equivocal in my view of the suitability of the original version of this manuscript. My concerns centered on two major points. First, the authors demonstrate no challenging example of how their method works, instead showing data only for cases in which the answer was already known. Demonstrating a novel protein-protein interaction method without using it to detect a novel interaction is troublesome, as much can happen when the answer is not known, most of it not good. Second, the lack of any useful quantification, especially by Western blot but even in terms of how to interpret ELISA results, is extremely problematic. I believe that my concern here is similar to the other referee's concerns about issues like expression levels, foldability of the fusions and effects of the orientation of the tags, and can be partly captured in the questions: How does a user of the method know if a signal for interaction is real? How much prior testing of fusions, use of standards, comparison to previous ELISA results, reversal of tag orientations, expression of the fusions at variable levels, etc. must be done to be confident that one is either detecting or not detecting an interaction?

In response to my concerns, the authors provide no substantive new experiment to address them. Instead, they seem to argue, at considerable length, that my concerns are unwarranted. Regarding their carrying out a single screen to demonstrate that the SIMPL method is able to detect a novel interaction, they write that this is beyond the scope of this manuscript. I respectfully disagree. If it takes three years of a new grant to demonstrate this application, then perhaps the method at this time is still early enough in its development that publication is premature.

Regarding the quantification of Western analysis, they write that "it is very difficult to set universal standards for Western analysis-based assays." This seems tantamount to saying that their assay results in Figures 1c, 1d, 1e, 1f, 2b, 4a, 4b, 4c and Supplementary Figures should be viewed with caution because they are not quantifiable.

The text still contains numerous worrisome rationales or qualitative comparisons, such as:
(l. 176) It should be noted that a basal splicing signal appeared....However, the corresponding rapamycin-treated sample shows a dramatically increased signal...
(l. 199) Although two combinations...showed relatively elevated basal splicing levels, these background signals did not interfere...
(l. 251) ...while no obvious signal was observed. Assay with the...construct exhibited a more marked splicing signal....We speculate that the signal enhancement is caused by...N-terminal tagging...
(l. 265) We reason that the high basal signal...derived...when its local concentration is high....Accumulated background splicing protein masked the signal...

I am sorry that my previous comment about Figure 3f was unclear, and see how this lack of clarity

might have led the authors to explain how ROC analysis works. I was asking if the authors could quantify, e.g. in a heatmap, how each of the interactions of the Positive Reference Set behaved in the SIMPL assay, rather than only indicating by a red box that the signal was "positive." However, they do now provide these data in Supplementary Figure 3.

My request for the ELISA results to be at least roughly calibrated to signals on the Western blots elicited the response that I should compare Figure 2b with 3C, but I do not believe that viewing these two panels adequately addresses this request.

My request for a clear statement of what constitutes "detection" of an interaction was addressed with the comment that I should see the response to Comment 3 (where I could not find the answer) and that "Additionally, like all screening methods, use of sophisticated statistical algorithms can greatly aid in hit identification and analysis."

Given that my major concerns from the original manuscript have not been resolved, I do not find that the revised manuscript is acceptable.

Point by Point Response to Reviewer #2

Comment 1. ...the authors demonstrate no challenging example of how their method works, instead showing data only for cases in which the answer was already known. Demonstrating a novel protein-protein interaction method without using it to detect a novel interaction is troublesome, as much can happen when the answer is not known, most of it not good. ... Regarding their carrying out a single screen to demonstrate that the SIMPL method is able to detect a novel interaction, they write that this is beyond the scope of this manuscript. I respectfully disagree. If it takes three years of a new grant to demonstrate this application, then perhaps the method at this time is still early enough in its development that publication is premature.

Response to comment 1: We thank for the reviewer's suggestion, understand the concerns, but respectfully disagree with the conclusion that "*perhaps the method at this time is still early enough in its development that publication is premature*". A major cause of failure of some methods in detecting new PPIs is due to the lack of a strict and thorough characterization of these methods. However, the full characterization of a new method is itself remarkably challenging. While screening for novel interactions is certainly a key potential use of SIMPL our goal here is to deal with the non-trivial issue of fully characterizing and defining the strengths and limitations of our new methodology, which we have done in great depth using a range of established assays and a comprehensive (and well-recognized) control set of protein interactions that was designed by others in the field for expressly this purpose. This sort of characterization is essential for any new technique to help researchers determine if and how they can adopt it to their work, while avoiding failure and significant waste in labour, time and expense. It is also how many other method developers first introduce their new techniques, some of which have also been previously published at Nature Communications (*Nat Communications* **8**, 2244; *Nature Communications* **8**, 1524; *Nature Communications* **6**, 7294; *Mol. Cell. Proteomics* **13**, 3332–3342; *Mol. Syst. Biol.* **14**, e8071).

With respect to established vs. novel interactions, in order to get an idea of how a method behaves one needs to run full benchmarking analysis using a diverse set of known control interactions; not all methods detect various PPIs equally, and the use of known controls is one of the best recognized ways to estimate how a particular technique performs. In terms of a statistical assessment, use of a defined set provides much more useful performance information than attempting to identify unknown interactions, which themselves then require further validation by other methods.

Regarding using SIMPL to screen novel interactions, while we appreciate the reviewer's opinion, based upon our extensive experience in proteomics, screening for novel interactions using any methodology represents an enormous amount of work, both in terms of running the screens themselves and then performing the necessary functional validations to establish the biological relevance of detected interactions. Our manuscript is already of considerable length and adding the large amount of data would completely distract from the major aim of this study. Note that screens using SIMPL are in fact underway in our group, however we look forward to presenting the results of these screens in future manuscripts where they can be given the proper focus they deserve.

Comment 2: ...the lack of any useful quantification, especially by Western blot but even in terms of how to interpret ELISA results, is extremely problematic. ... Regarding the quantification of Western analysis, they write that "it is very difficult to set universal standards for Western analysis-based assays." This seems tantamount to saying that their assay results in

Figures 1c, 1d, 1e, 1f, 2b, 4a, 4b, 4c and Supplementary Figures should be viewed with caution because they are not quantifiable. ... My request for the ELISA results to be at least roughly calibrated to signals on the Western blots elicited the response that I should compare Figure 2b with 3C, but I do not believe that viewing these two panels adequately addresses this request.

Response to Comment 2: We thank for the reviewer's suggestions and feel sorry for not having incorporated the suggested quantitative analyses in our previous revision. These analyses, including new experimental data, are now added in the new manuscript as follow. **(1)** We quantified the splicing signals in Western analysis by measuring the densities of spliced protein (FRB-FKBP) bands in the dose response experiment (Fig. 1c) and the time course experiment (Fig. 1d) of rapamycin-induced FRB/FKBP interaction. It should be noted that the profile of splicing in dose response is consistent with that obtained with ELISA (Fig. 3c). **(2)** We quantified the splicing signals in Western analysis of rapamycin-induced FRB/FKBP interaction in various SIMPL formats (Fig. 2b). The quantitative signals are also consistent with those obtained by ELISA analysis. **(3)** We selected 40 PPIs from the PRS, including 10 pairs with high ELISA signal and 10 pairs with low ELISA signal in either IN/IC or IN/CIC format, and re-tested them by Western blot analysis. Quantification of the Western bands showed excellent correlation of the two methods.

Comment 3: *I believe that my concern here is similar to the other referee's concerns about issues like expression levels, foldability of the fusions and effects of the orientation of the tags, and can be partly captured in the questions: How does a user of the method know if a signal for interaction is real? How much prior testing of fusions, use of standards, comparison to previous ELISA results, reversal of tag orientations, expression of the fusions at variable levels, etc. must be done to be confident that one is either detecting or not detecting an interaction?*

Response to comment 3: These questions represent fundamental challenges associated with the use of any interactive proteomics technology and are one of the major reasons that proper validation and characterization of new assays using established controls/methods (i.e. the entire focus of our current manuscript) are critical. Most of these concerns have been more expansively addressed in detail in our rebuttal letter and previously revised manuscript, and the first reviewer has accepted our explanation. Regarding the concern with tagging, it should be based on previous knowledge of the functionality of the target protein termini. If it is unknown, assays can be performed using all possible tagging configurations. This is common practice in many proteomics assays and, for example, is the same strategy that is used by nanoBRET (*ACS Chem. Biol.* **10**, 1797–1804).

Comment 4: *The text still contains numerous worrisome rationales or qualitative comparisons, such as:*

(l. 176) It should be noted that a basal splicing signal appeared....However, the corresponding rapamycin-treated sample shows a dramatically increased signal...

(l. 199) Although two combinations...showed relatively elevated basal splicing levels, these background signals did not interfere...

(l. 251) ...while no obvious signal was observed. Assay with the...construct exhibited a more marked splicing signal....We speculate that the signal enhancement is caused by...N-terminal tagging...

(l. 265) We reason that the high basal signal...derived...when its local concentration is high....Accumulated background splicing protein masked the signal...

Response to comment 4: We feel sorry for our unclear description in the previous version. We have modified them with quantitative descriptions and highlighted the modification with red in the new manuscript.

Comment 5: *I was asking if the authors could quantify, e.g. in a heatmap, how each of the interactions of the Positive Reference Set behaved in the SIMPL assay, rather than only indicating by a red box that the signal was “positive.”*

Response to comment 5: According to this suggestion, we have created a heatmap and present it as Supplementary Figure 3b in the revised manuscript. Please note that the cutoff values are also labelled.

Comment 6: *My request for a clear statement of what constitutes “detection” of an interaction was addressed with the comment that I should see the response to Comment 3 (where I could not find the answer) and that “Additionally, like all screening methods, use of sophisticated statistical algorithms can greatly aid in hit identification and analysis.”*

Response to comment 6: We feel sorry for not having made clear explanation. A “detection”, or “positive”, indicates a protein pair with its SIMPL signal above the cutoff value. However, like in any method, a positive can be either true or false. Therefore we made an efforts to increase the rate of true positive as measured with PRS and to decrease the rate of false positive as measured with RRS, in parallel to reducing false negative and enhancing true negative. That is the reason we performed strict benchmarking characterization. We feel that we have addressed the reviewer’s concern in the Response to Comment 3 in our previous Response. The misunderstanding is probably derived from the reviewer’s desire to set a universal bar for all different experiments. However, this is not practical in either SIMPL or most PPI methods due to the inherent complexities and variation of PPIs and PPI methods. This is also the reason various statistical algorithms and analysis methods are often employed to interpret proteomics data. In the case of SIMPL, we normalize the signal to bait expression. Currently we have not included the measurement error of bait and the measurement of prey in our analysis. New analysis algorithms including these factors could further increase the precision of PPI measurement and will be something we investigate in the future as we continue to develop and make practical use of the SIMPL platform.

Reviewers' Comments:

Reviewer #2:

Remarks to the Author:

This is a re-revised manuscript that now deals with a few of the concerns of the original review, rather than pointing out that those concerns were unwarranted. The revision represents at least an initial effort to establish some degree of a quantitative basis for detecting interactions. In particular, some of the Western signals have now been quantified and at least in one case have been compared to ELISA results. However, I am still puzzled by how a user establishes that an interaction has been detected, even if you accept the authors' contention that the cutoff value can be different for different protein pairs. Indeed, on line 204, the authors cite "a robust 1.5 fold increase" in spliced signal after rapamycin treatment as evidence of interaction detection. Lines 278-281, however, indicate the lack of interaction detection of a different pair because "the signal was not significantly enhanced after EGF stimulation" given the density ratio was 1:1.7. Perhaps these contradictory statements provide some sense of what is creating confusion about these data.

Point-by-Point Response to Reviewer #2

Comment 1: *This is a re-revised manuscript that now deals with a few of the concerns of the original review, rather than pointing out that those concerns were unwarranted. The revision represents at least an initial effort to establish some degree of a quantitative basis for detecting interactions. In particular, some of the Western signals have now been quantified and at least in one case have been compared to ELISA results.*

Response to comment 1: In response to the reviewer's previous comment, we indeed made the comparison in two cases. The first is the dose response of rapamycin-induced FRB/FKBP1A interaction with the ELISA data shown in Fig.3c and quantified Western blot in Fig. 1d. Since only one format (IN/IC) was tested in Fig. 1c, we complemented the comparison by quantifying the Western blots for all four formats (Fig. 2b) although in this experiment only one rapamycin dose was tested. Both ELISA and Western data demonstrate good consistency. Secondly, we selected 44 protein pairs in reference sets and performed Western analysis (Supplementary Fig. c,d). The quantification similarly exhibits consistency.

Comment 2: *However, I am still puzzled by how a user establishes that an interaction has been detected, even if you accept the authors' contention that the cutoff value can be different for different protein pairs. Indeed, on line 204, the authors cite "a robust 1.5 fold increase" in spliced signal after rapamycin treatment as evidence of interaction detection. Lines 278-281, however, indicate the lack of interaction detection of a different pair because "the signal was not significantly enhanced after EGF stimulation" given the density ratio was 1:1.7. Perhaps these contradictory statements provide some sense of what is creating confusion about these data.*

Response to Comment 2: We would like to clarify that "a robust 1.5 fold increase" was observed in ELISA analysis and the 1:1.7 ratio was observed in Western blot analysis. Since these are different types of experiment, performed under two different conditions on two PPIs with distinct biological functions, we believe they are not comparable. Western blot analysis is less quantitative because too many factors can affect its quantitative features such as detection method, exposure time and quantification method. We believe that is why Nature Communications adopts the policy that "Quantitative comparisons between samples on different gels/blots are discouraged". More importantly, ELISA is characterized with a much wider dynamic range than Western analysis. Thus, identical or similar values obtained from different methods do not have the same meaning.